# Morc3 silences endogenous retroviruses by enabling Daxx-mediated histone H3.3 incorporation

Sophia Groh [1], Anna Viktoria Milton [1], Lisa Katherina Marinelli [1], Cara V. Sickinger [1], Angela Russo [1], Heike Bollig [1], Gustavo Pereira de Almeida[1], Andreas Schmidt[2], Ignasi Forné[2], Axel Imhof [2] & Gunnar Schotta [1✉]

Endogenous retroviruses (ERVs) comprise a significant portion of mammalian genomes. Although specific ERV loci feature regulatory roles for host gene expression, most ERV integrations are transcriptionally repressed by Setdb1-mediated H3K9me3 and DNA methylation. However, the protein network which regulates the deposition of these chromatin modifications is still incompletely understood. Here, we perform a genome-wide single guide RNA (sgRNA) screen for genes involved in ERV silencing and identify the GHKL ATPase protein Morc3 as a top-scoring hit. Morc3 knock-out (ko) cells display de-repression, reduced H3K9me3, and increased chromatin accessibility of distinct ERV families. We find that the Morc3 ATPase cycle and Morc3 SUMOylation are important for ERV chromatin regulation. Proteomic analyses reveal that Morc3 mutant proteins fail to interact with the histone H3.3 chaperone Daxx. This interaction depends on Morc3 SUMOylation and Daxx SUMO binding. Notably, in Morc3 ko cells, we observe strongly reduced histone H3.3 on Morc3 binding sites. Thus, our data demonstrate Morc3 as a critical regulator of Daxx-mediated histone H3.3 incorporation to ERV regions.

---

[1] Division of Molecular Biology, Biomedical Center (BMC), Faculty of Medicine, Ludwig-Maximilians-University (LMU) Munich, Großhaderner Straße 9, 82152 Martinsried, Germany. [2] Protein Analysis Unit, Biomedical Center (BMC), Faculty of Medicine, Ludwig-Maximilians-University (LMU) Munich, Großhaderner Straße 9, 82152 Martinsried, Germany. ✉email: gunnar.schotta@bmc.med.lmu.de

Endogenous retroviruses compose a significant portion of mammalian genomes. During evolution, most ERV integrations in mammals were highly mutated or partially deleted and are thus unable to generate functional retroviral particles. However, remnant ERVs harbor binding sites for transcription factors and can act as regulatory elements to drive host gene expression[1–6]. This physiological role for the regulation of normal development is contrasted by pathological effects of aberrant ERV regulation in neurological diseases[7] and cancer[8–10]. Many ERV families are thus transcriptionally repressed by a silencing pathway that involves the formation of H3K9me3 heterochromatin[11,12]. The current model for H3K9me3 establishment on ERVs is based on sequence-specific binding of KRAB-ZnF proteins, which recruit the co-repressor Trim28 and the histone methyltransferase Setdb1[12–14]. The establishment of DNA methylation is likely to happen during early embryogenesis, and maintenance of high methylation levels on ERVs is later ensured by Uhrf1/Dnmt1[15,16]. However, recent data indicate that Dnmt1/Uhrf1 can also induce de novo DNA methylation on ERVs[17]. Full establishment of H3K9me3 and DNA methylation, as well as reduced chromatin accessibility on ERVs, require activities, such as histone H3.3 deposition by Atrx/Daxx[18–21], chromatin remodeling by Smarcad1[22], and chromatin assembly by Chaf1a/b[23]. In ES cells, chromatin remodeling activities enforce a dynamic exchange of Histone H3.3[24]. In this context, histone H3.3 deposition is necessary to replace evicted nucleosomes and ensure low chromatin accessibility[24]. Currently, it is not clear how histone H3.3 turnover is regulated and coordinated with other chromatin-modifying activities to restrict chromatin accessibility and to mediate heterochromatin spreading on ERVs.

We have previously identified a small heterochromatin inducing sequence (SHIN) in Intracisternal A Type particle (IAP) elements[20]. The SHIN sequence is a 160 bp region from the GAG coding sequence in IAPEz elements. Genomic insertions of reporters containing the SHIN sequence attract Setdb1-dependent H3K9me3[20]. Using a genome-wide single-guide RNA (sgRNA) screen for genes involved in SHIN silencing we now identified Morc3 as a player in ERV silencing. Morc3 is a GHKL type ATPase protein that can form a closed dimer in the ATP-bound state[25]. It further contains a CW-type zinc finger domain that negatively regulates ATPase activity[26]. The binding of ligands, such as histone H3K4me3 or influenza virus protein NS1 peptides, to the CW domain, relieves suppression of ATPase activity and could regulate the turnover of ATPase domain-mediated dimerization and opening[27,28]. Although H3K4me3-mediated interaction of Morc3 with promoter regions in mouse ES cells was reported[25], the molecular roles of Morc3 in transcriptional regulation or chromatin organization are currently unclear. We demonstrate that Morc3 binds ERV sequences and that loss of Morc3 results in increased chromatin accessibility, reduced H3K9me3, and de-repression of ERVs. We detect an interaction of Morc3 with the histone H3.3 chaperone Daxx, which depends on the Morc3-ATPase cycle and SUMOylation. This interaction is crucial for Daxx-mediated histone H3.3 incorporation as Morc3 ko ES cells lose H3.3 on ERVs. Thus, our data indicate Morc3 as a regulator of Daxx-mediated histone H3.3 incorporation.

## Results

### Identification of Morc3 as an ERV silencing factor.
To identify factors regulating IAPEz silencing we performed a genome-wide sgRNA screen based on the integration of an IAP SHIN reporter in mouse ES cells (T90 cells, Fig. 1a). This reporter contains a doxycycline-inducible promoter driving the expression of EGFP and a zeocin resistance gene. In the wild-type condition, doxycycline induction results in poor reporter activation due to its heterochromatic state, whereas impaired SHIN silencing allows

doxycycline-induced reporter activity[20]. We used a pooled genome-wide lentiviral sgRNA library[29] to transduce T90 ES cells, followed by doxycycline induction to activate the reporter locus. Reporter activity also results in zeocin resistance. Therefore, we applied zeocin selection to detect cells with an activated reporter resulting from impaired heterochromatin. As a control, we collected a second pool of cells without selection pressure. We then identified the sgRNAs which were enriched in the zeocin selected cells (Fig. 1b, Supplementary Data 1). Top enriched sgRNAs targeted the major known ERV silencing factors, such as Dnmt1, Uhrf1, Setdb1, Trim28, and Atrx/Daxx (Fig. 1b). Another top hit was Morc3, a protein not previously implicated in ERV silencing. Due to the mode of selection with zeocin, we also identified genes that might be related to zeocin resistance, DNA damage repair, and apoptosis (Supplementary Data 1). We compared our data with a similar screen for IAP silencing factors which was recently published[30]. Overall, we found very little overlap between the two screens. Only the major silencing factors, such as Atrx, Daxx, Dnmt1, Uhrf1, Trim28, and our top hit Morc3 were present in both datasets (Supplementary Fig. S1a). Different silencing inducing sequences (SHIN sequence vs. 5′ LTR-UTR) used in the respective reporters could be responsible for the differences in the screening hits.

To validate selected screening hits, we performed SHIN silencing maintenance assays for candidate gene knock-outs (Fig. 1c). Control sgRNA treatment did not result in an activatable SHIN reporter, whereas sgRNA knock-out of Setdb1 and Morc3 resulted in significant SHIN de-repression (Fig. 1d, FACS gating strategy in Supplementary Fig. S1b). We further tested additional screening candidates in the SHIN maintenance assay and could partially confirm our screening results (Fig. 1e). We then complemented the functional testing with a SHIN initiation silencing assay in which we test whether a newly introduced SHIN reporter with a constitutively active promoter can be silenced in different genetic backgrounds (Fig. 1f). Here, the results differed from the maintenance assay as the DNA methylation pathway appeared less important for establishing silencing, while Setdb1/Trim28 and Atrx/Daxx were still critical for silencing (Fig. 1g). Since Morc3 was important in both maintenance and initiation of silencing we decided to functionally characterize this protein in more detail.

### Morc3 binds different families of endogenous retroviruses.
To assess if Morc3 directly regulates ERV silencing we performed ChIP-seq experiments using a knock-in cell line that expresses 3xFLAG tagged Morc3 at endogenous levels (Supplementary Fig. S2). We identified 2737 peaks that were shared between at least two out of three replicates (Supplementary Data 2). Annotation with genomic features revealed that most Morc3 peaks associate with ERVs (Fig. 2a). We then classified the ERV families to which Morc3 peaks associate and found a major enrichment with IAPEz, LTRIS2, and RLTR families (Fig. 2b). Due to the lack of polymorphisms, many ERV integrations cannot be precisely mapped. To better assess the association of Morc3 with ERV families we used RepEnrich2[31] to categorize all mappable Morc3 ChIP-seq reads into ERV families. This analysis confirmed the prominent association of Morc3 with IAP, RLTR, and LTRIS families (Fig. 2c, Supplementary Data 3). The most prominent ERV family bound by Morc3 are IAPEz elements, from which the SHIN sequence used in our screen originates. We generated cumulative coverage plots for IAPEz elements to determine whether Morc3 displays preferential binding with distinct IAPEz features. Interestingly, comparison of the Morc3 profile with ChIP-seq profiles for Setdb1, Trim28, and H3K9me3 (Fig. 2d) revealed prominent co-enrichment of the silencing factors with

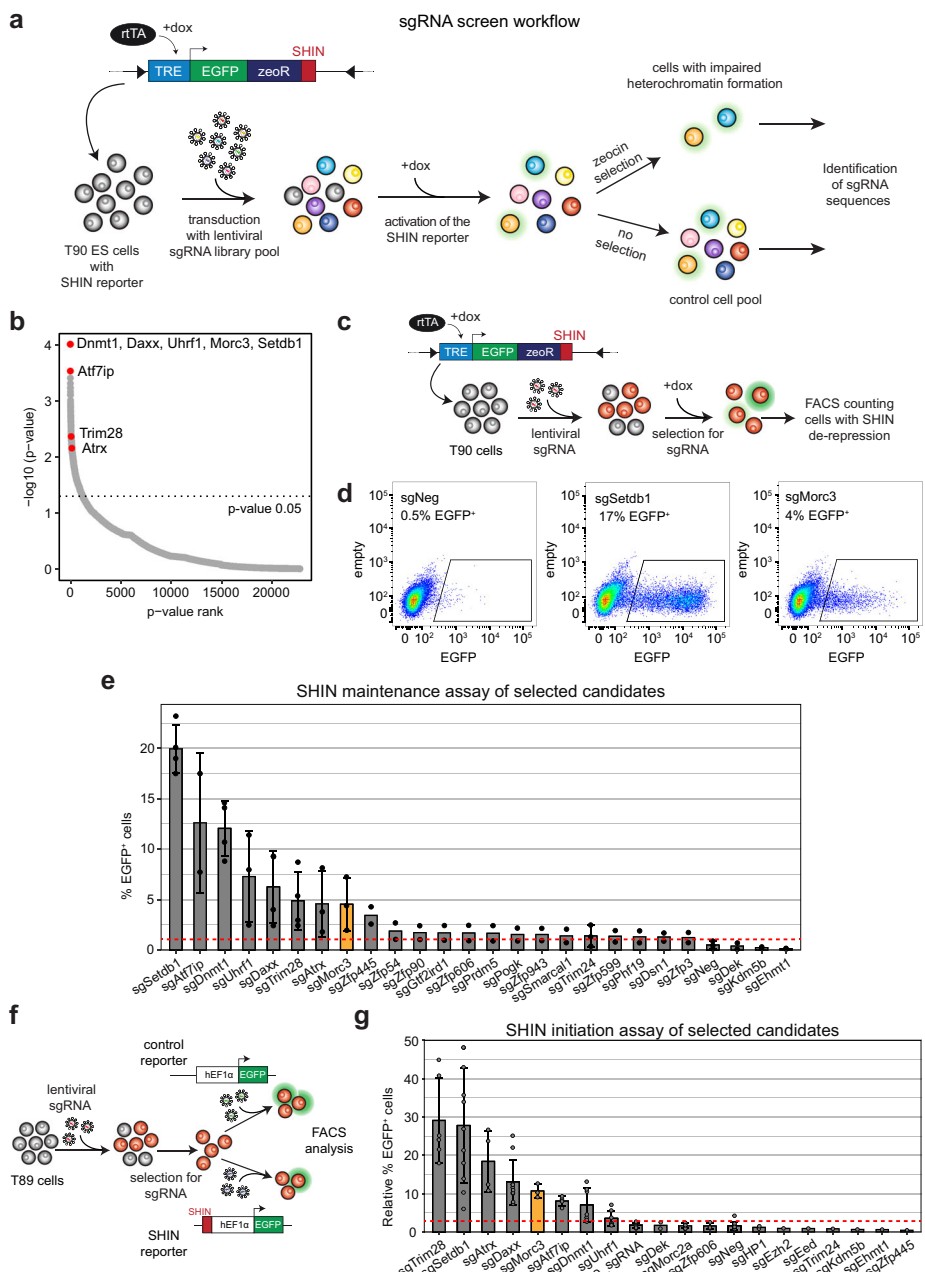

**Fig. 1 A genome-wide sgRNA screen for SHIN silencing identifies Morc3. a** Setup of the sgRNA screen. T90 ES cells containing an inducible SHIN reporter[20] were transduced with a genome-wide sgRNA library pool. Cells were selected for sgRNA vectors and the reporter was activated with doxycycline. Two independent cell batches were cultured to harvest non-selected cells representing the control, and zeocin-resistant cells representing cells with impaired heterochromatin on the SHIN reporter. **b** Dot plot showing the sgRNA screen results ordered by *p*-value rank (RIGER SecondBestRank scoring). Major ERV silencing factors are indicated. Morc3 represents a top hit in the screen. **c** Schematic of the SHIN silencing maintenance assay. T90 ES cells were transduced with sgRNAs for selected candidates and treated for integration with puromycin. Subsequently, doxycycline was added to the cells to induce reporter activity. EGFP expression was analyzed by FACS. **d** FACS plots depicting EGFP fluorescence in cells with activated SHIN reporter. Almost no activity was detected in cells transduced with a control sgRNA, demonstrating full SHIN silencing. sgRNAs targeting *Setdb1* or *Morc3* result in SHIN de-repression as indicated by cells showing EGFP expression. **e** Bar plot depicting the results of the SHIN maintenance silencing assay with selected candidate genes. The red dotted line indicates the background reporter activity. Data are presented as mean values +/− SD of biological replicates per sgRNA (*n* = 2–4, for details see "Statistics and reproducibility" section). **f** Schematic of the SHIN silencing initiation assay. T89 ES cells were transduced with sgRNAs for selected candidates and treated for integration with puromycin. Subsequently, the cells were transduced with a control virus without the SHIN sequence or the SHIN reporter with a constitutive promoter, respectively. EGFP expression was analyzed by FACS and the percentage of EGFP expressing cells, relative to the control reporter was calculated. **g** Bar plot depicting the results of the SHIN initiation silencing assay with selected candidate genes. The red dotted line indicates the background reporter activity. Data are presented as mean +/− SD of biological replicates per sgRNA (*n* = 1–12, for details see "Statistics and reproducibility" section).

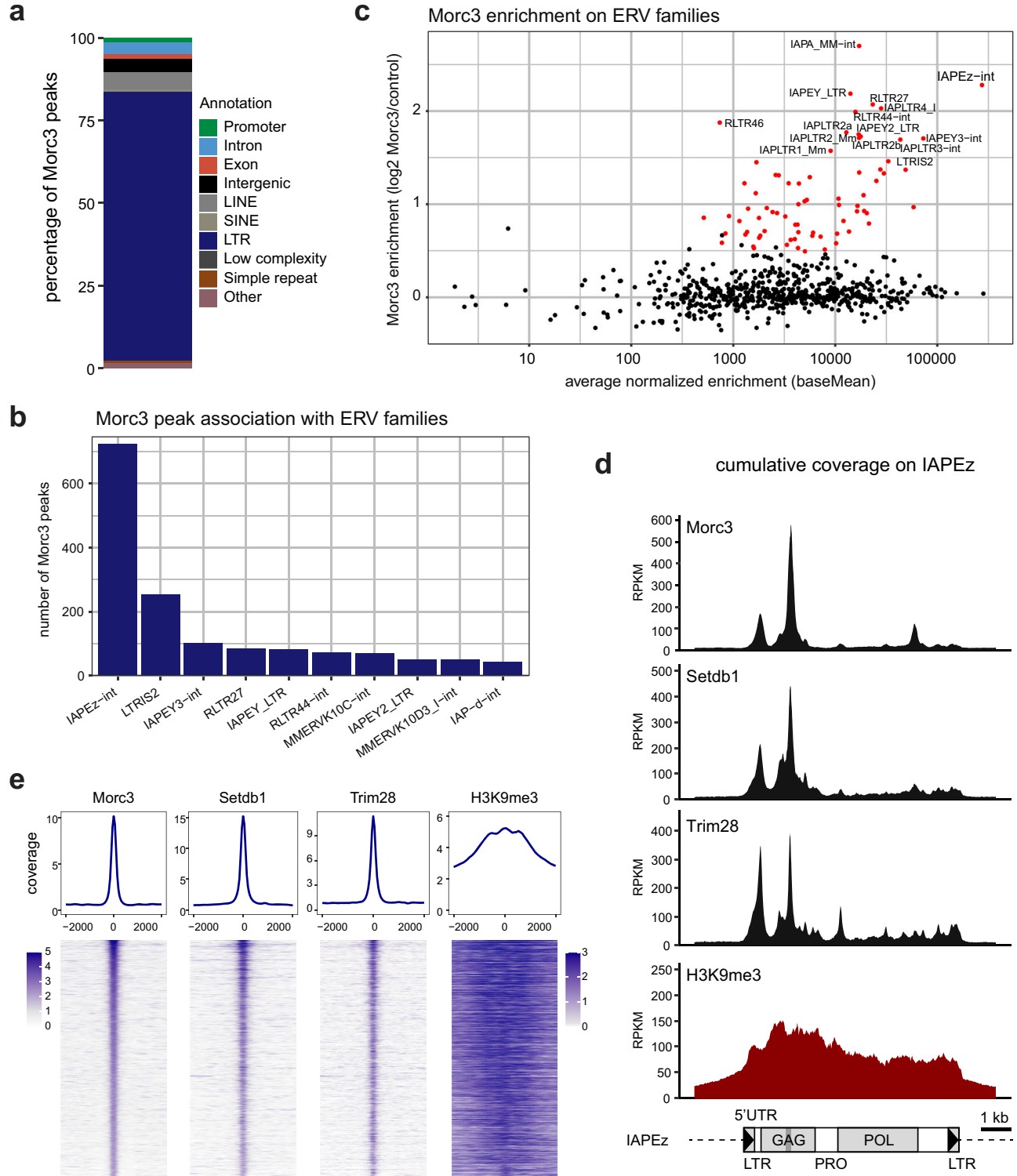

**Fig. 2 Morc3 is enriched on different families of endogenous retroviruses. a** Genomic features associated with the 2737 Morc3 peaks. **b** Morc3 peak association with ERV families. The bar plot shows the number of Morc3 peaks associating with the ten most enriched ERV families. **c** Dot plot showing Morc3 ChIP enrichment on ERV families. Red dots highlight ERV families with significant Morc3 over background enrichment. Selected ERV families are labeled. **d** Cumulative coverage plot of Morc3, Setdb1, Trim28, and H3K9me3 ChIP-seq profiles on IAPEz elements. Prominent enrichment is over the 5′ UTR and the GAG region. The position of the SHIN sequence is indicated as a dark gray bar. Setdb1 ChIP-seq is from a 3xFLAG knock-in ES cell line. Trim28 ChIP-seq data are from ref. [23]. **e** Morc3 peaks are associated with Setdb1 and Trim28 binding and feature high H3K9me3. Top panel: Density plot showing the average occupancy of Morc3, Setdb1, Trim28, and H3K9me3 on Morc3 peaks. Lower panel: Read-density heat map showing the normalized coverage of Morc3, Setdb1, Trim28, and H3K9me3 on Morc3 binding sites. Distance from the peak center is given in bp. Setdb1 ChIP-seq is from a 3xFLAG knock-in ES cell line. Trim28 ChIP-seq data are from ref. [23].

the IAP-GAG region containing the SHIN sequence and with the 5′UTR region that is also able to induce reporter silencing[14]. The similarity of the Morc3 binding pattern to Setdb1 and Trim28 suggests a functional interplay with the H3K9me3 pathway (Fig. 2d). To investigate whether Morc3 more generally associates with Setdb1 and Trim28 on its identified peaks, we generated a read-density heatmap of Setdb1, Trim28, and H3K9me3 on Morc3 peaks (Fig. 2e). This analysis also revealed prominent enrichment of Setdb1, and Trim28 on Morc3 peaks, further supporting Morc3 association with H3K9me3 repressed heterochromatin.

**Morc3 knock-out results in ERV de-repression**. To investigate the role of Morc3 in ERV silencing we generated Morc3 knock-out ES cells (Supplementary Fig. S3a). Transcriptome analysis by RNA-seq revealed significant dysregulation of 252 genes (adjusted *p*-value < 0.01) with a trend towards genes being upregulated (Fig. 3a, Supplementary Data 4). We then asked if Morc3 could be directly involved in the regulation of these genes by determining Morc3 peaks in the vicinity of their transcription start sites. We found that 64 upregulated and 18 downregulated genes had Morc3 peaks in <100 kb distance to their promoter (Supplementary Data 5). When we investigated the detailed peak annotation of these Morc3 peaks we found a strong association with ERV LTR sequences. The highest enriched ERV family was LRTIS2, which was found on 25 Morc3 peaks associated with upregulated genes (Supplementary Data 5). These data suggest that genes are indirectly regulated through de-repressed ERV LTRs which could act as enhancer elements for neighboring genes. Three prominent examples for upregulated genes are shown in Fig. 3B. *Irak3* has two Morc3 peaks in its gene body which associate with RLTR13B2 and ORR1A3, respectively. *Ube2l6* features Morc3 binding in an LTRIS2 element, overlapping the 3′UTR region. *Cd200* shows Morc3 binding with an LTRIS2 element downstream of its gene body. To attribute the effects on gene regulation to Morc3 function, we generated rescue ES cell lines by transducing the full-length Morc3 cDNA into Morc3 ko ES cells (Supplementary Fig. S3b). RNA-seq analysis of Morc3 rescue cell lines revealed largely normalized gene expression compared to Morc3 ko cells (Supplementary Fig. S4a). We validated selected Morc3 target genes by RT-qPCR in wild type, Morc3 ko, and rescue cells. All selected target genes displayed de-repression in Morc3 ko, whereas expression was reduced in rescue cells (Fig. 3c).

To determine Morc3 roles in ERV repression we performed SHIN silencing initiation assays. Morc3 knock-out cells displayed impaired SHIN silencing, whereas rescue cells can efficiently establish SHIN repression (Fig. 3d). We also generated Morc3 knock-out cells from T90 SHIN reporter ES cells to investigate Morc3-dependent silencing maintenance (Supplementary Fig. S3c). Without doxycycline induction, no EGFP expression could be detected in wild-type or Morc3 ko cells (Fig. 3e). We then monitored EGFP expression from the SHIN reporter with 2 and 4 days of doxycycline induction. Wild-type cells failed to activate the reporter and almost no EGFP positive cells could be detected (Fig. 3e). In contrast, silencing of the SHIN reporter was strongly impaired in Morc3 ko cells (Fig. 3e), demonstrating that Morc3 is important for the maintenance of SHIN silencing.

To determine if IAP and potentially other ERV families are de-repressed in Morc3 ko cells, we counted RNA-seq reads corresponding to ERVs with RepEnrich2 and calculated differentially expressed ERV families (Fig. 3f, Supplementary Data 6). We found that indeed, major Morc3 target families, such as IAP, RLTR, and LTRIS displayed significant de-repression, which can be restored in Morc3 rescue cells (Fig. 3f, Supplementary

Fig. S4b). The extent of regulation is greatly reduced compared to Setdb1 knock-out where IAP de-repression is around 10–100-fold[13,32]. Therefore, our data suggest that Morc3 is a contributing factor for ERV silencing that acts redundantly with other mechanisms in the context of ERVs. However, individual ERV integrations, probably with less redundancy in silencing mechanisms, are strongly de-repressed in Morc3 ko ES cells and affect the expression of neighboring genes.

**Morc3 knock-out leads to reduced H3K9me3 and increased chromatin accessibility**. To characterize the chromatin changes coinciding with Morc3 loss, we measured H3K9me3 in wild type, Morc3 ko, and rescue cells. Visual inspection of H3K9me3 profiles on Morc3 peaks revealed different patterns of changes. Therefore, we clustered Morc3 peaks according to changes in H3K9me3 patterns and generated a read-density heatmap of Morc3 peak clusters with H3K9me3 pattern in wild-type cells, together with fold change representation in Morc3 ko and rescue cells (Fig. 4a). We found that H3K9me3 was generally reduced in cluster I. Clusters II and III displayed a biased reduction of H3K9me3 towards one side of the peak center. Clusters IV and V showed reduced H3K9me3 mainly in the peak center and cluster VI did not show appreciable changes in H3K9me3. Overall, the average H3K9me3 signal on Morc3 peak centers is significantly reduced in Morc3 ko cells and fully restored in Morc3 rescue cells (Fig. 4b). The genomic features underlying the different Morc3 peak clusters cannot easily explain the different patterns of change as all clusters display prominent enrichment of LTR elements (Supplementary Fig. S5a).

H3K9me3 heterochromatin is characterized by low chromatin accessibility which could prevent efficient access of transcriptional activators[33,34]. To test if chromatin accessibility changes in Morc3 ko cells, we performed ATAC-seq experiments in wild type, Morc3 ko, and rescue ES cells. Altogether we identified 444 peaks with significantly higher accessibility in Morc3 ko ES cells, while only 10 peaks were less accessible (Fig. 4c). Increased chromatin accessibility is directly related to Morc3 loss as we found reduced chromatin accessibility of these peaks in Morc3 rescue cells (Fig. 4c). Only a subset of the regulated ATAC-seq peaks was directly overlapping with Morc3 peaks, suggesting direct and indirect effects of Morc3 on chromatin accessibility (Fig. 4c).

We tested if enhanced chromatin accessibility in Morc3 ko could relate to activation of repressed enhancers by performing H3K27ac ChIP-seq. We observed a subset of Morc3 peaks which showed increased H3K27ac in Morc3 ko cells (Fig. 4d). These regions appeared to associate with low levels of H3K27ac already in wild-type cells (Supplementary Fig. S5b). We then asked for the overlap of H3K9me3 reduction, H3K27ac increase, and higher ATAC-seq accessibility in Morc3 ko cells. This analysis revealed that almost all peaks that displayed signs of enhancer activity (higher H3K27ac/ATAC-seq accessibility) displayed reduced H3K9me3 (Fig. 4e). The genomic features of these Morc3 peaks are characterized by ERV integrations and putative ENCODE enhancers (Supplementary Fig. S5c–e), supporting the view that Morc3 helps to repress distinct developmental enhancers. An example of a Morc3 repressed enhancer is shown in the genome browser screenshot for *Irak3* (Fig. 4f). This locus contains two Morc3 peaks in H3K9me3 chromatin. The peak distal to the TSS overlaps with an RLTR13B2 element and does not display chromatin changes in Morc3 ko cells. The second, promoter-proximal peak overlaps with an ORR1A3 element, but also with a region of developmental enhancers. This region shows strongly increased H3K27ac, ATAC-seq signals, and one-sided loss of H3K9me3 signal in Morc3 ko cells (Fig. 4f). Examples of

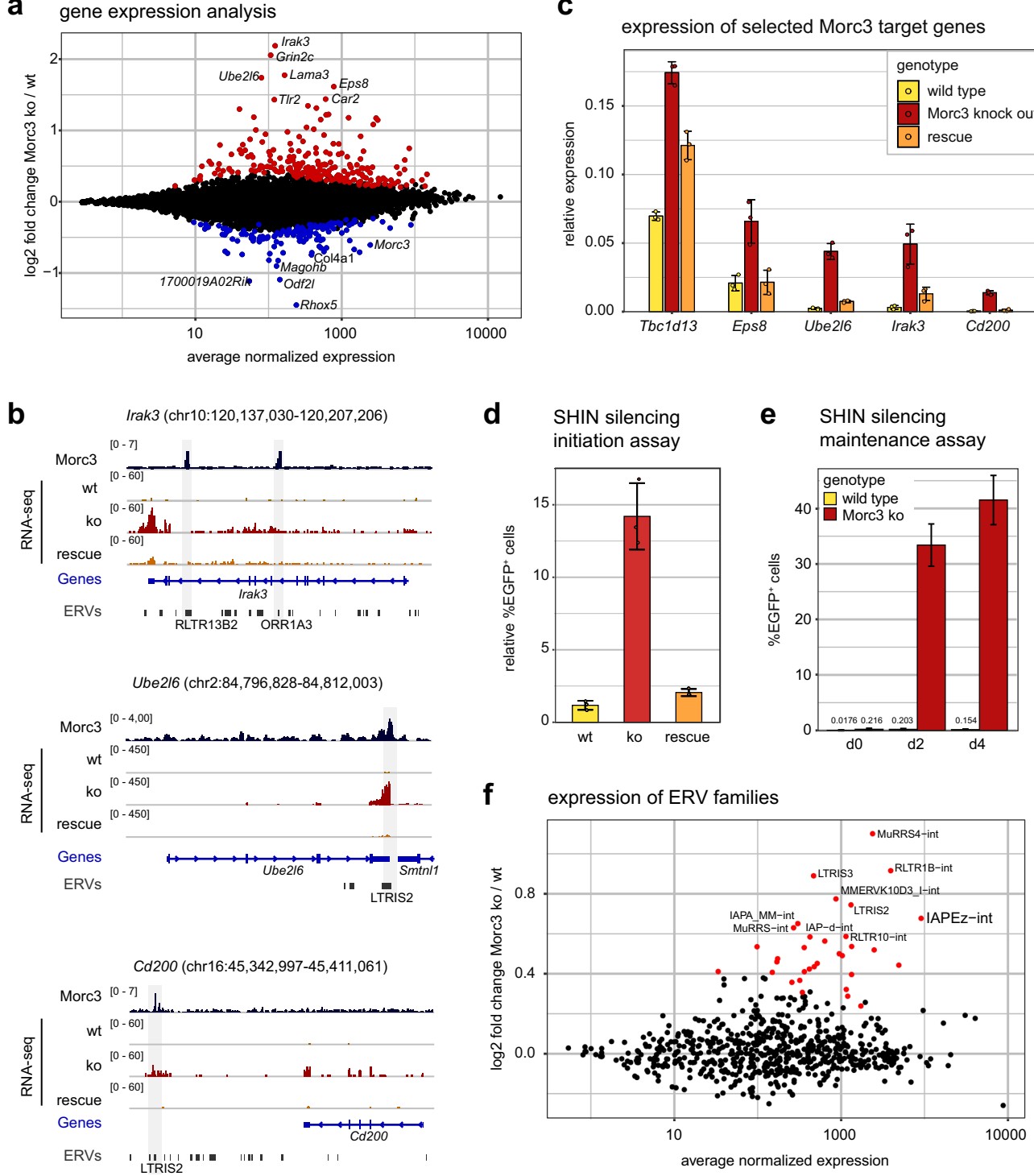

Morc3 repressed enhancers associated with ERV (LTRIS2) integrations are in the vicinity of Morc3 target genes *Cd200* and *Ube2l6* (Supplementary Fig. S6).

Since IAPEz elements represent the major target of Morc3 peaks, we calculated the difference in H3K9me3, ATAC, and H3K27ac signals comparing Morc3 ko vs. wild type and rescue vs. wild-type cells (Supplementary Fig. S5f). We found that H3K9me3 is decreased in the IAP-GAG region, concomitant with increased chromatin accessibility, while H3K27ac could not be detected (Supplementary Fig. S5f, top panels). These chromatin changes were reverted in Morc3 rescue cells (Supplementary Fig. S5f, lower

panel). We extended this analysis using RepEnrich2 analysis and detected reduced H3K9me3 and increased chromatin accessibility on other ERV families (Supplementary Fig. S7a, b).

Together, our data demonstrate that Morc3 is critical for the full maintenance of H3K9me3 and low chromatin accessibility on distinct ERV families and distinct developmental enhancers.

**Morc3-dependent heterochromatin requires a functional Morc3-ATPase cycle and -SUMOylation.** Next, we thought to investigate the importance of conserved Morc3 domains and

**Fig. 3 Morc3 knock-out cells display de-repression of genes and distinct ERV families. a** Dot plot showing average expression vs. log2-fold change of coding genes in wild type vs. Morc3 knock-out ES cells. Colored dots indicate genes with significantly changed expression (DESeq2 adjusted *p*-value < 0.05, *n* = 3 biological replicates for each condition). Positions of relevant genes are indicated. **b** Genome browser view of Morc3-dependent expression changes on selected target genes (*Irak3*, *Ube2l6*, and *Cd200*). Morc3 peaks are located on ERVs within or in close vicinity to these genes. Transcriptional upregulation of the target genes in Morc3 ko cells can be rescued by expression of wild-type Morc3. **c** RT-qPCR analysis of selected Morc3 target genes in wild type, Morc3 ko, and Morc3 rescue ES cells. Bar graph depicts mean relative expression to control genes (*Actin* and *Hprt*). Error bars indicate the standard deviation of replicate experiments (*n* = 3). Individual data points are shown as colored dots. **d** SHIN initiation silencing assay in wild type, Morc3 ko, and Morc3 rescue ES cells. Bar graph depicts the mean relative percentage of EGFP positive cells of SHIN-reporter transduced cells relative to control virus transduced cells. Error bars indicate the standard deviation of replicate experiments (*n* = 3). Individual data points are shown as colored dots. **e** SHIN maintenance silencing assay in wild type and Morc3 ko ES cells. Bar graph depicts the mean percentage of EGFP positive cells after doxycycline induction for 2 and 4 days, respectively. Error bars indicate the standard error of replicate experiments (*n* = 3). **f** Dot plot showing average expression vs. log2-fold change of ERV families in wild type vs. Morc3 knock-out ES cells. Colored dots indicate ERVs with significantly changed expression (DESeq2 adjusted *p*-value < 0.05, *n* = 3 for each condition). Positions of relevant ERV families are indicated.

SUMOylation for its role in ERV heterochromatin establishment. Morc3 has an N-terminal GHKL ATPase domain which undergoes conformational changes coupled with ATP binding and hydrolysis. The Morc3-ATPase domain is monomeric in the ATP unbound state and dimerizes upon ATP binding[25]. ATP hydrolysis and dissolution of the ATPase domain dimer are negatively regulated by the CW domain. Ligand binding of the CW domain relieves this inhibition and results in higher ATPase activity[28]. To test the function of the Morc3-ATPase cycle (Supplementary Fig. S8a) and SUMOylation for ERV silencing we generated distinct Morc3 mutant rescue cell lines (Fig. 5a). The ATP hydrolysis mutant (E35A) and the ATP binding mutant (G101A) should disrupt the ATPase cycle due to the inability to bind or hydrolyze ATP[35]. The ΔN mutant is unable to dimerize and displays reduced ATPase activity[28]. The CW ligand binding mutant (W419A) has a reduced ATPase activity and would result in a slowed-down ATPase cycle[28]. The 5KR mutant cannot be SUMOylated[35] and potentially affects SUMO-dependent protein interactions. The Morc3 mutant rescue cell lines display slight overexpression of Morc3 (Supplementary Fig. S8b), which is not directly related to the amount of Morc3 protein. In particular, the ATP hydrolysis mutant shows low Morc3 protein levels, which could suggest reduced stability of this mutant (Supplementary Fig. S8c). All mutant rescue cell lines display a similar nuclear distribution of Morc3 (Supplementary Fig. S8d). ChIP-seq analyses revealed concordant localization of most Morc3 mutant proteins with Morc3 peaks and IAPEz elements (Fig. 5b, Supplementary Fig. S8e). The ATP hydrolysis mutant failed to properly localize to Morc3 targets and the ATP binding mutant showed reduced binding to Morc3 binding sites (Fig. 5b, Supplementary Fig. S8e). In the ChIP-seq data of the rescue cells, we noticed subtle binding of Morc3 to H3K4me3 promoter regions which was reported before[25]. A more systematic analysis revealed that, although ChIP-seq of Morc3 in the 3xFLAG knock-in cells did not show association with H3K4me3, in wild type and some mutant rescue cell lines Morc3 could be detected on H3K4me3 promoters (Supplementary Fig. S9a). The CW mutant failed to localize to these regions, which would support H3K4me3-dependent recruitment of Morc3 through the CW domain[25]. Impaired H3K4me3 localization of the SUMO mutant suggests that SUMOylation contributes to H3K4me3 binding (Supplementary Fig. S9a). We did not observe changes in promoter H3K4me3 in Morc3 knock-out cells (Supplementary Fig. S9b) and we did not detect large-scale changes in transcription (see Fig. 3a). Therefore, we think that the major functional targets of Morc3 are ERV sequences and distinct enhancers.

We then functionally tested the Morc3 mutant rescue cells in SHIN initiation silencing assays and found that all mutant proteins failed to induce SHIN silencing (Fig. 5c). We also performed RT-qPCR analysis for selected Morc3 target genes

and ERVs and observed de-repression, comparable to Morc3 knock-out cells (Fig. 5d, Supplementary Fig. S10a). To assess chromatin changes, we performed ATAC-seq and H3K9me3 ChIP-seq experiments in the mutant rescue cell lines. All mutants displayed increased accessibility on ATAC-seq peaks of ERV families (Fig. 5e). In line with increased chromatin accessibility, we detected reduced H3K9me3 on Morc3 peaks (Fig. 5f). Differences in cumulative ATAC-seq and H3K9me3 ChIP-seq coverage on IAPEz elements demonstrated compromised chromatin architecture in Morc3 mutant rescue cells (Supplementary Fig. S10b, c). Consistent with the de-repression of Morc3 target genes we detected increased chromatin accessibility on Morc3-associated peaks (Supplementary Figs. S10d and S11). Taken together our data demonstrate that a fully functional ATPase cycle and Morc3 SUMOylation are critical for Morc3 functionality.

**The Morc3-ATPase cycle and SUMOylation are needed for its interaction with Daxx.** Conformational changes during the Morc3-ATPase cycle, as well as SUMOylation, might affect interactions with other proteins that could be related to ERV silencing. To test this hypothesis, we determined the protein interaction context of Morc3 using ChIP-mass spectrometry analysis. This approach uses cross-linking and solubilization of chromatin through sonication. Therefore, proteins detected in this approach represent direct and stable protein interactions as well as low-affinity interactions which are more transient in nature, and proteins for which the interaction with Morc3 is mediated through DNA/RNA fragments. We found 489 proteins significantly enriched in the Morc3 ChIP-MS data (Fig. 6a, Supplementary Data 7). Importantly, we detected the major ERV silencing factors Setdb1, Trim28, Atrx/Daxx, and Dnmt1/Uhrf1. We also observed a strong association with SUMO proteins, corresponding to high SUMOylation of Morc3[35] and consistent with the role of SUMO in ERV repression[23]. The functional categorization of Morc3 interactors using Panther protein class enrichment analysis revealed that many proteins in the Morc3 context belong to chromatin and histone-modifying activities (Fig. 6b). Interestingly a large proportion of proteins related to RNA binding and RNA processing, suggesting that Morc3 may have roles in RNA metabolism.

Next, we investigated if the protein interaction context would change in the Morc3 mutant rescue cell lines. In this analysis, we did not include the ATP hydrolysis mutant (E35A) due to impaired recruitment to Morc3 peaks. Comparison between ChIP-MS results between wild type and mutant Morc3 proteins did not reveal large overall changes, suggesting that the interaction context is largely preserved in the mutants (Fig. 6c). However, focusing on the major ERV silencing factors, we found a selective reduction of Daxx in ChIP-MS data of most Morc3

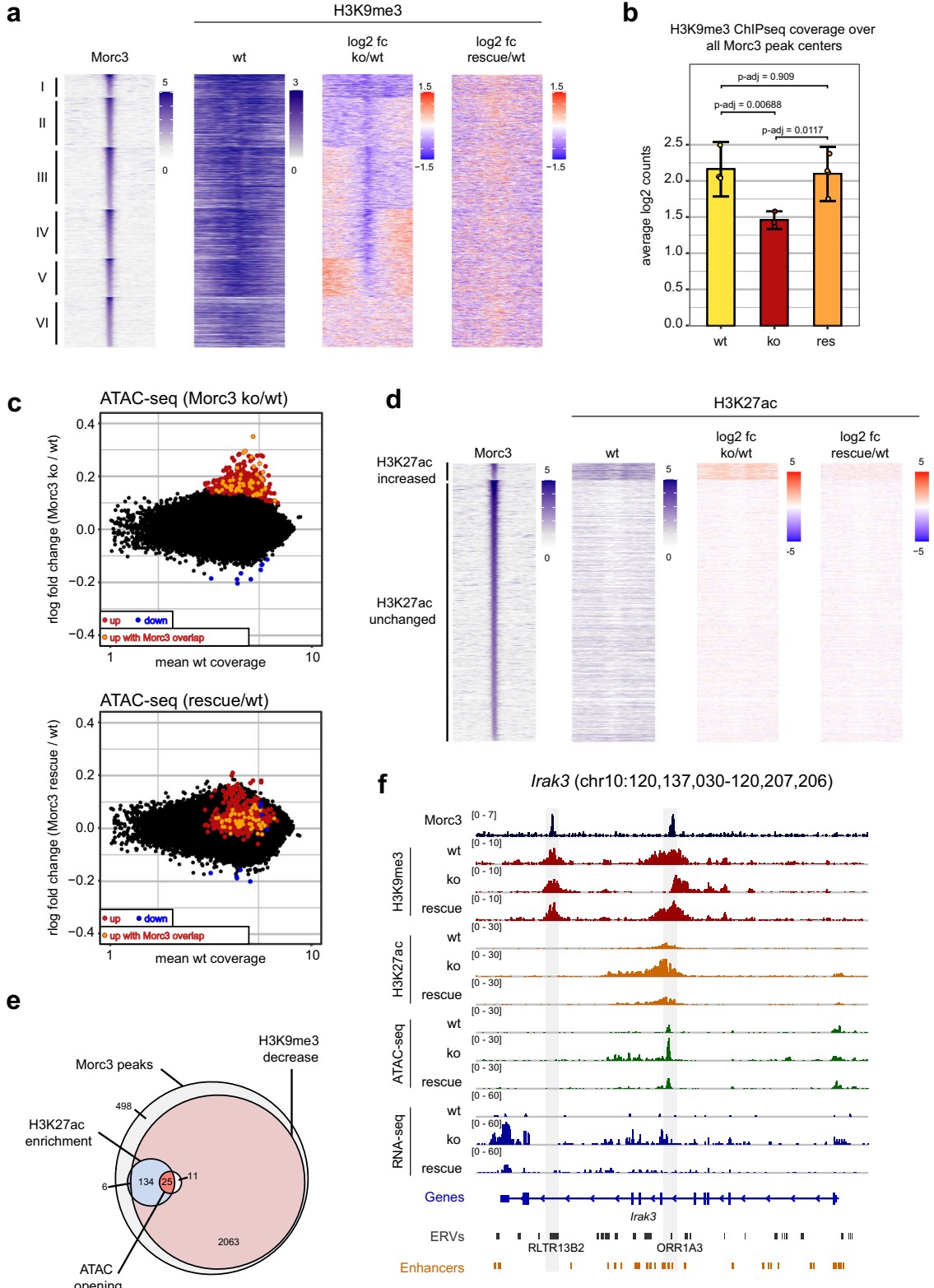

mutant proteins (Fig. 6c). Only in the Morc3 CW mutant Daxx association was minimally impaired (Fig. 6c). Reduced association with Daxx cannot be attributed to its reduced abundance, since nuclear Daxx levels are maintained in the rescue cell lines (Supplementary Fig. S12). We validated interaction with Daxx by co-immunoprecipitation with wild-type Morc3 (Fig. 6d). In agreement with the ChIP-MS data, Morc3 mutant proteins failed to co-immunoprecipitate Daxx (Fig. 6d). These data indicate that

a fully functional Morc3-ATPase cycle, as well as Morc3 SUMOylation, are needed for Daxx interaction.

Daxx is known to regulate ERV silencing through a C-terminal SUMO interaction motif (SIM)[20]. Since Morc3 is highly SUMOy-lated and Daxx failed to interact with the Morc3 SUMOylation mutant, we set out to determine if the interaction between Morc3 and Daxx is mediated through this C-terminal SIM. We performed ChIP-MS experiments comparing Daxx knock-out cell lines with re-

**Fig. 4 Morc3 knock-out leads to reduced H3K9me3 and increased chromatin accessibility on target regions. a** Changes in H3K9me3 on Morc3 peaks. Morc3 peaks were grouped in six clusters according to changes in H3K9me3 in Morc3 ko ES cells. Read-density heatmaps show the normalized coverage of Morc3 and H3K9me3 on Morc3 peaks, the log2-fold change in H3K9me3 signal between Morc3 ko and wild type ES cells, and the log2-fold change in H3K9me3 between Morc3 rescue and wild type cells. Depletion of H3K9me3 is colored in blue, increased H3K9me3 appears red. **b** H3K9me3 coverage on Morc3 peaks. Bar graph depicts mean normalized reads over all Morc3 peak centers (100 bp bin). Error bars indicate the standard error of replicate experiments ($n = 3$). Individual data points are shown as colored dots. $p$-values were calculated by a Tukey multiple comparisons of means Anova test with 95% family-wise confidence level. **c** Dot plot showing average coverage vs. log2-fold change of ATAC peaks in wild type vs. Morc3 knock-out ES cells (top panel) and wild type vs. Morc3 rescue ES cells (lower panel). Colored dots indicate ATAC peaks with significantly increased (red) or decreased (blue) coverage in Morc3 ko ES cells (DESeq2 adjusted $p$-value < 0.05, $n = 3$ for each condition), peaks with significantly increased ATAC coverage and overlapping with Morc3 peaks are marked in orange. **d** H3K27ac coverage on Morc3 peaks. Morc3 peaks were divided into 2 groups according to the change of H3K27ac in Morc3 ko compared to wt ES cells. Read-density heatmaps show the normalized coverage of Morc3 and H3K27ac on Morc3 peaks, the log2-fold change in H3K27ac signal between Morc3 ko and wild-type ES cells, and between Morc3 rescue and wild-type cells. Depletion of H3K27ac is colored in blue, increased H3K27ac appears red. **e** Venn diagram summarizing Morc3-dependent chromatin changes. **f** Genome browser view of Morc3-dependent chromatin changes on *Irak3*. Positions of two Morc3 peaks are indicated by gray boxes. The promoter-proximal Morc3 binding site displays selective loss of H3K9me3, increased H3K27ac, and increased chromatin accessibility, concomitant with elevated transcription.

expression of wild-type Daxx protein or Daxx$^{\Delta SIM}$. Lack of the C-terminal SIM reduced association with SUMO proteins and resulted in strongly reduced binding with Morc3 (Fig. 6e). In addition, wild-type Daxx could co-immunoprecipitate Morc3, whereas Daxx$^{\Delta SIM}$ failed to bind Morc3 (Fig. 6f). Together, our data indicate that SUMOylated Morc3 interacts with Daxx through the C-terminal SIM.

**Morc3 is important for Daxx-mediated histone H3.3 incorporation.** Recent studies have shown the role of Daxx-mediated H3.3 incorporation in ERV silencing and suggest a dynamic turnover of histones in ERV heterochromatin[18,24,36]. Therefore, we asked if the association of Morc3 with Daxx is critical for histone H3.3 incorporation on Morc3 target sites. We found histone H3.3 prominently enriched on Morc3 peak regions (Fig. 7a, Supplementary Fig. S13a). Histone H3.3 was largely lost in Morc3 ko cells and could be re-established in Morc3 wild-type rescue cells (Fig. 7a). Consistent with impaired Daxx interaction, all Morc3 mutants failed to re-establish histone H3.3 on Morc3 peak regions (Fig. 7a). On Morc3 target genes, we detected changed patterns of histone H3.3 which might be linked with the transcriptional activation of these loci (Fig. 7b, Supplementary Fig. S14). For example, *Irak3* showed slightly increased histone H3.3 incorporation in a broad region close to the promoter-proximal peak (Fig. 7b), which is associated with increased chromatin accessibility and H3K27ac (see Fig. 4f). The distal Morc3 peak, in contrast completely lost histone H3.3 association (Fig. 7b). Cumulative coverage of histone H3.3 confirmed that Morc3 peaks with repressed chromatin almost completely lost histone H3.3, whereas Morc3 peaks with gained H3K27ac signal could maintain significant levels of this histone variant (Fig. 7c), which might be linked with Daxx-independent histone H3.3 deposition in the context of enhancer activation[37,38].

Then, we investigated histone H3.3 changes on ERV chromatin. We found that Morc3 target ERV families displayed strongly reduced histone H3.3 incorporation in Morc3 ko cells, which could be rescued with wild-type Morc3 (Fig. 7d). Cumulative coverage of histone H3.3 on IAPEz elements revealed a strong association in wild-type cells and almost complete loss in Morc3 knock-out cells (Fig. 7e). Re-expression of Morc3 could rescue histone H3.3 deposition on IAPEz elements (Fig. 7e), whereas Morc3 mutant proteins failed to rescue histone H3.3 deposition (Supplementary Fig. S13b).

Finally, we thought to compare the transcriptional changes observed in Morc3 knock-out cells with transcriptional changes in Daxx or histone H3.3 knock-out cells using published data for knock-out vs rescue cells[39]. Interestingly, a subset of Morc3 regulated ERV families displayed a trend towards being

upregulated in Daxx and histone H3.3-ko cells (Supplementary Fig. S15A). In addition, several Morc3 target genes were upregulated in Daxx and histone H3.3-ko cells (Supplementary Figs. S15b and S16).

Together, our data demonstrate that Morc3 is critical for Daxx-mediated histone H3.3 deposition on distinct ERV families and enhancers.

## Discussion
Our data show that Morc3 is a critical regulator of ERV chromatin in mouse ES cells. Other members of the MORC protein family have already been implicated in ERV regulation. Morc1 was found to regulate transposable elements in the mouse germline[40] and Morc2a regulates LINE1 repression in mouse ES cells[41]. In human cells, MORC2 was identified to influence HUSH complex silencing of HERVs[42,43]. Although the MORC family of proteins represent important regulators of ERV silencing in different cell types, their mechanism of action was largely obscure. We can show a specific function of Morc3 in regulating Daxx-mediated histone H3.3 incorporation to maintain ERV heterochromatin (Fig. 7F). We found that in absence of Morc3, ERV heterochromatin loses H3K9me3, shows increased accessibility and reduced levels of histone H3.3. A previous study has demonstrated high histone H3.3 turnover on ERV regions[24]. Smarcad1 evicts nucleosomes from ERV heterochromatin and, Daxx-dependent replacement by histone H3.3-containing nucleosomes is needed to maintain low chromatin accessibility. Loss of histone H3.3 or Daxx, therefore, leads to more accessible chromatin and less H3K9me3 on ERVs. We found that Morc3 knock-out cells display a very comparable phenotype, supporting the notion that Morc3 function is needed for proper Daxx activity. Based on our data we propose the following model for this process. High nucleosome turnover on ERV heterochromatin is induced by chromatin remodelers, such as Smarcad1. The re-establishment of these nucleosomes is mediated by Daxx, which contributes histone H3.3/H4 dimers. For this activity, Daxx needs to interact with SUMOylated Morc3 through its SUMO interaction motif. Morc3 undergoes an ATPase cycle with conformational changes that could be necessary for Daxx interaction and may function as "licensing step" for Daxx-mediated histone H3.3 deposition. In agreement with this model, we found that (I) Morc3 knock-out cells display impaired histone H3.3 deposition, (II) Daxx requires the C-terminal SIM domain to interact with Morc3, (III) the Morc3 mutant which cannot be SUMOylated failed to interact with Daxx (IV) Morc3 mutant proteins with impaired ATPase cycle fail to interact with Daxx. It is interesting to note that an efficient ATPase cycle is not only critical for

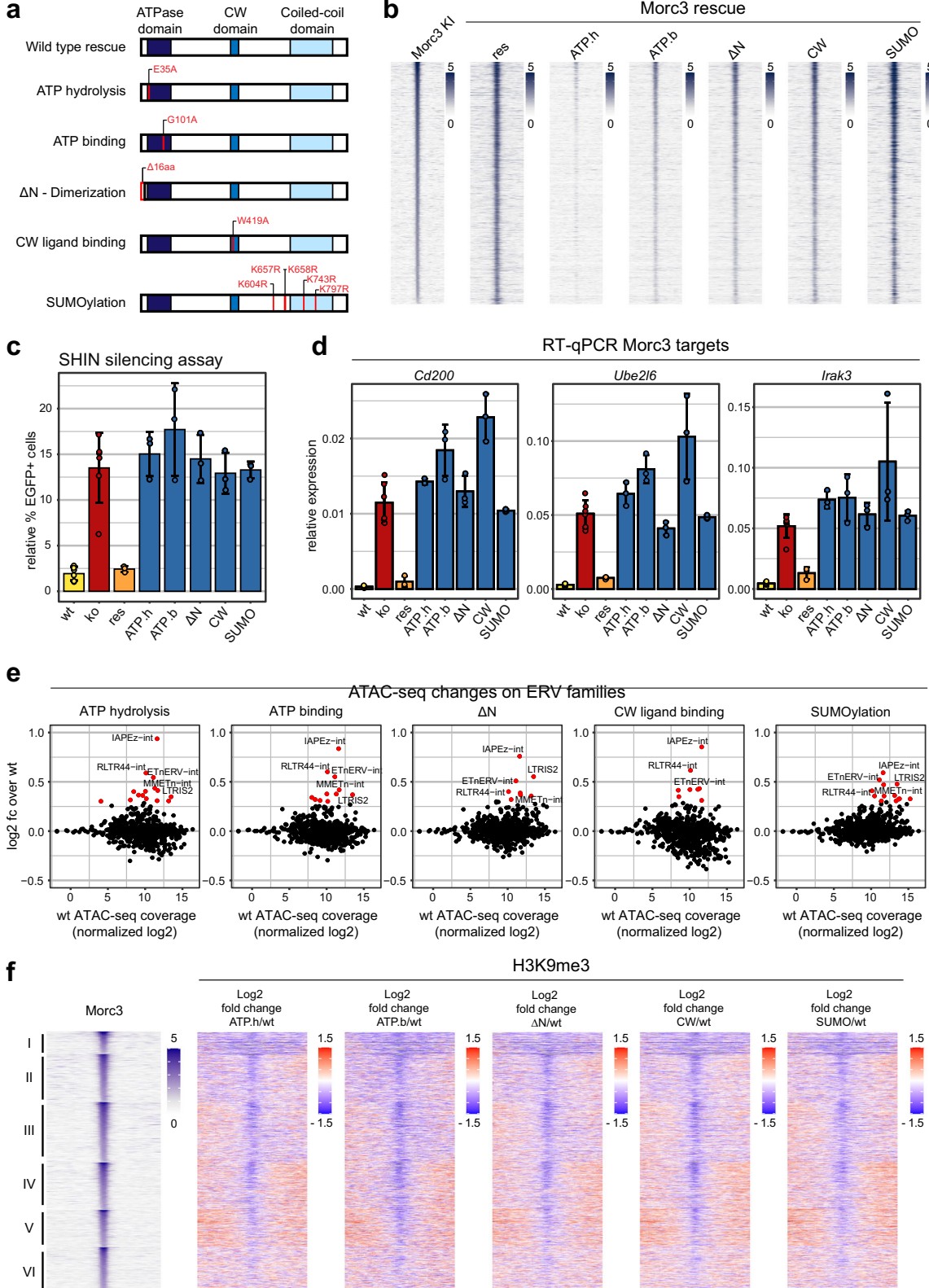

Morc3 function but also affects the silencing activity of MORC2[44].

The ATPase cycle is differently affected in the Morc3 mutant proteins. The Morc3-ATPase mutant protein is modeled after an Hsp90 mutant which binds ATP but fails to hydrolyze it[45]. We found that this mutant does not properly localize to Morc3 peak regions and, overall protein abundance is reduced which could

suggest a higher degradation rate. The ATP binding mutant is expected to impair dimerization and ATP hydrolysis. This mutant localizes to Morc3 peaks, but with overall less intensity as compared to wild type. The ΔN mutant does not dimerize and displays reduced ATPase activity[28]. Localization to Morc3 peaks is largely intact, but failure to re-establish histone H3.3 and ERV heterochromatin demonstrates that Morc3 dimerization and/or

**Fig. 5 Morc3-dependent heterochromatin requires a functional Morc3-ATPase cycle and SUMOylation. a** Morc3 mutations which impair the ATPase cycle or Morc3 SUMOylation. Five mutant constructs were generated to impair ATP binding, ATP hydrolysis, dimerization, CW ligand interaction, and SUMOylation. **b** Most Morc3 mutant proteins bind to Morc3 peaks. Read-density heat map showing the normalized coverage of Morc3 wild type and mutant proteins on Morc3 binding sites. The Morc3 ATP hydrolysis mutant displays strongly reduced coverage on Morc3 peaks. **c** SHIN initiation silencing assay with Morc3 mutant rescue cell lines. Bar graph depicts mean relative percentage of EGFP positive cells of SHIN-reporter transduced cells relative to control virus transduced cells. Error bars indicate standard deviation of replicate experiments ($n = 3$). Individual data points are shown as colored dots. Data for wild type, ko, and rescue are from Fig. 3d for comparison. **d** RT-qPCR analysis of selected Morc3 target genes in Morc3 mutant rescue ES cells. Bar graphs depict mean relative expression to control genes (*Actin* and *Hprt*). Error bars indicate standard deviation of replicate experiments ($n = 3$). Individual data points are shown as colored dots. Data for wild type, ko, and rescue are from Fig. 3c for comparison. **e** Dot plot showing average wt ATAC-seq coverage vs. log2-fold change of ERV families in wild type vs. Morc3 mutant rescue ES cells (ATP hydrolysis mutant, ATP binding mutant, ΔN, CW ligand binding, SUMOylation mutant). Colored dots indicate ERV families with significantly increased (red dots) or decreased (blue dots) coverage in Morc3 mutant rescue ES cells (adjusted *p*-value < 0.05, $n = 2$ for each mutant condition and $n = 3$ for wt). **f** Morc3 mutant proteins fail to rescue H3K9me3 on Morc3 binding sites. Morc3 peaks group in six clusters according to changes in H3K9me3 in Morc3 ko ES cells (compare with Fig. 4a). Read-density heatmaps show the fold change in H3K9me3 between Morc3 mutant rescue and wild-type ES cells. Depletion of H3K9me3 is colored in blue, increased H3K9me3 appears red. All mutants fail to rescue the ko phenotype.

ATP hydrolysis are needed for Daxx-mediated histone H3.3 incorporation. The effect of the CW mutant is more complex. The CW domain acts as a negative regulator of ATPase activity and requires ligand binding to relieve auto-inhibition. Thus, Morc3 is expected to feature full ATPase activity only in the context of CW ligand binding. Mutation of the CW domain impairs ligand binding and therefore results in reduced ATPase activity, while dimerization of the ATPase domain is not affected. The loss-of-function phenotype of the Morc3 CW mutant would therefore suggest that ligand (H3K4me3) binding, and probably efficient ATPase activity are needed for Morc3 function in vivo.

All Morc3 mutant proteins fail to establish histone H3.3 on Morc3 binding sites. Although ΔN, CW, and SUMO mutants can properly localize to chromatin, they show impaired interaction with Daxx. This is most prominent in co-IP experiments, where only relatively stable interactions are captured. In ChIP-MS we observed differences between these mutants, the SUMO mutant completely impaired association, whereas ΔN showed strongly reduced and CW only slightly reduced Daxx association. This could suggest that Daxx can localize to Morc3 binding sites with ΔN and CW mutants, but stable interaction with Morc3 and histone H3.3 deposition might then require an efficient ATPase cycle. Unfortunately, we failed to ChIP Daxx and were unable to distinguish whether Daxx binding to Morc3 peaks, or activity on its binding sites, was impaired in Morc3 knock-out and mutant rescue cell lines.

Our data show that histone H3.3 deposition is not generally critical for ERV silencing. Several ERV families that are bound by Morc3 and lose H3.3 do not show strong transcriptional upregulation. However, individual ERV integrations and distinct developmental enhancers strongly respond to Morc3 loss and can affect the regulation of genes in their vicinity. It will thus be important to explore the role of Morc3 for histone H3.3 deposition during differentiation and development, to assess if other members of the MORC family also regulate Daxx function and to understand if the dysregulation of their targets may contribute to the different diseases associated with mutations in MORC family members[42,46–48].

## Methods

**Cell culture**. Feeder-independent ES cells were cultured in ES cell medium (500 ml high glucose DMEM (Sigma, D6429), 91 ml (15%) fetal bovine serum (Sigma, F7524), 6.05 ml penicillin-streptomycin (Sigma, P4333), 6.05 ml MEM non-essential amino acid solution (Sigma, M7145), 1.2 ml 0.35% 2-mercaptoethanol (Sigma M7522), and 2.4 ml homemade LIF) on 10 and 15 cm gelatin-coated plates.

**Lentivirus production**. Lentiviral particles were produced to perform Daxx rescue, to carry reporter plasmids, to introduce sgRNAs for CRISPR-Cas9 knock-out, and to deliver the screening libraries. 293T cells were transfected with a mix of 24 µg

plasmid DNA consisting of 8 µg lentiviral transfer vector, 8 µg of each of the packaging plasmids psPAX2 (#183) and pLP-eco-env (#811). All plasmids are listed in Supplementary Table S1. The DNA was mixed with 120 µl 2.5 M CaCl$_2$ and adjusted to a final volume of 1200 µl with H$_2$O. Subsequently, 1200 µl 2 × HBS solution (50 mM HEPES, 280 mM NaCl, 1.5 mM Na$_2$HPO$_4$, adjusted to pH 7.05 with NaOH) was added slowly and dropwise to the mix while vortexing. The transfection mix was added immediately to the 293T cells seeded 1 day ahead at a density of 4 million per 10 cm dish. Four to eight hours after transfection, the medium that contained precipitates of calcium phosphate and DNA was removed, cells were washed with 1 × PBS and fresh medium was added. The virus-containing culture supernatant was harvested 48 h after transfection.

**Lentiviral transduction of mouse ES cells by spinoculation**. For lentiviral transduction, 200 thousand ES cells per well were seeded on gelatinized 6-well dishes in ES medium containing 8 µg/ml polybrene (Sigma, #H9268). Viral supernatant was added to the cells and plates were spun in a prewarmed centrifuge at 1000 × g for 1 h at 37 °C to enhance viral transduction by improved viscosity for infection. After centrifugation, the medium was carefully replaced or diluted with fresh ES medium to reduce toxic side effects of polybrene.

**sgRNA screen**. The genome-wide sgRNA screen was conducted using the 2-vector system (lentiGuide-Puro) GeCKOv2 pooled library. The plasmid library was obtained from the Zhang lab (Addgene catalog #1000000053) and amplified in *E. coli*[29]. To validate the complexity of the library the purified plasmid pool was sequenced. Plasmids were used for lentivirus production by calcium phosphate transfection of 293T cells. To ensure complexity of the library ten parallel virus preparations with a total amount of 80 µg plasmid DNA were set up. Harvested supernatants were pooled. To identify optimal virus concentration for achieving a multiplicity of infection (MOI) of 0.3, virus titers were determined for each virus lot. Infectivity was tested by transducing ES cells with the GeCKOv2 library through spinoculation of 300 thousand cells per well in 6-wells with different volumes of virus supernatant (between 1 and 100 µl). After 48 h, cells were transferred to 15 cm plates and put under selection with 0.5 µg/ml puromycin until colonies were detected. The screening experiment was performed using T90 cells, which are ES cells carrying the TetO-EGFP-T2A-Zeo-GAG2.22 reporter at one specific integration[20]. T90 cells also carry the rtTA for doxycycline induction of the TRE promoter and a Cas9 transgene. For saturation, we aimed for infection of 100 cells per sgRNA. As the libraries carry 65.000 different sgRNAs, at least 6.5 million ES cells ought to be transduced. As the MOI should be below 30%, 30 million T90 cells were transduced with the lentiviral sgRNA library pool at an MOI between 27–28%. Transduced cells were selected by puromycin treatment after 2 days with 0.5 µg/ml puromycin. The SHIN reporter was activated 2 days after transduction with 0.1 µg/ml doxycycline. To determine the distribution of sgRNAs without selection pressure, the cell pool was divided into two groups. The control pool was cultured in medium containing 0.5 µg/ml puromycin and 0.1 µg/ml doxycycline to determine the baseline sgRNA distribution. The selection pool was treated with either 25 or 50 µg/ml zeocin from day 4 on. All cells were passaged every 2 days and harvested 8 days after transduction. For the control pool, 20 million cells, and for the zeocin-treated pool 400,000 cells were collected. Extraction of genomic DNA was performed using the DNeasy Blood and Tissue Kit Mini (Qiagen, #69504). DNA was sheared using a syringe and 27 × g needle, and DNA concentration was determined with the Q-bit fluorometer (Invitrogen). Quantitative PCR (qPCR) was used on genomic DNA extracted from ES cells in the library screen to determine the number of cycles necessary for amplification of the sgRNA sequences with oligonucleotides GS3369 and GS3371 (Supplementary Table S7). PCR was carried out with the Fast SYBR® Green Master Mix™ (Applied Biosystems) in a LightCycler480™ (Roche). The reactions were performed in a total volume of 20 µl in a 384-well plate (Sarstedt). Ct values were generated by the

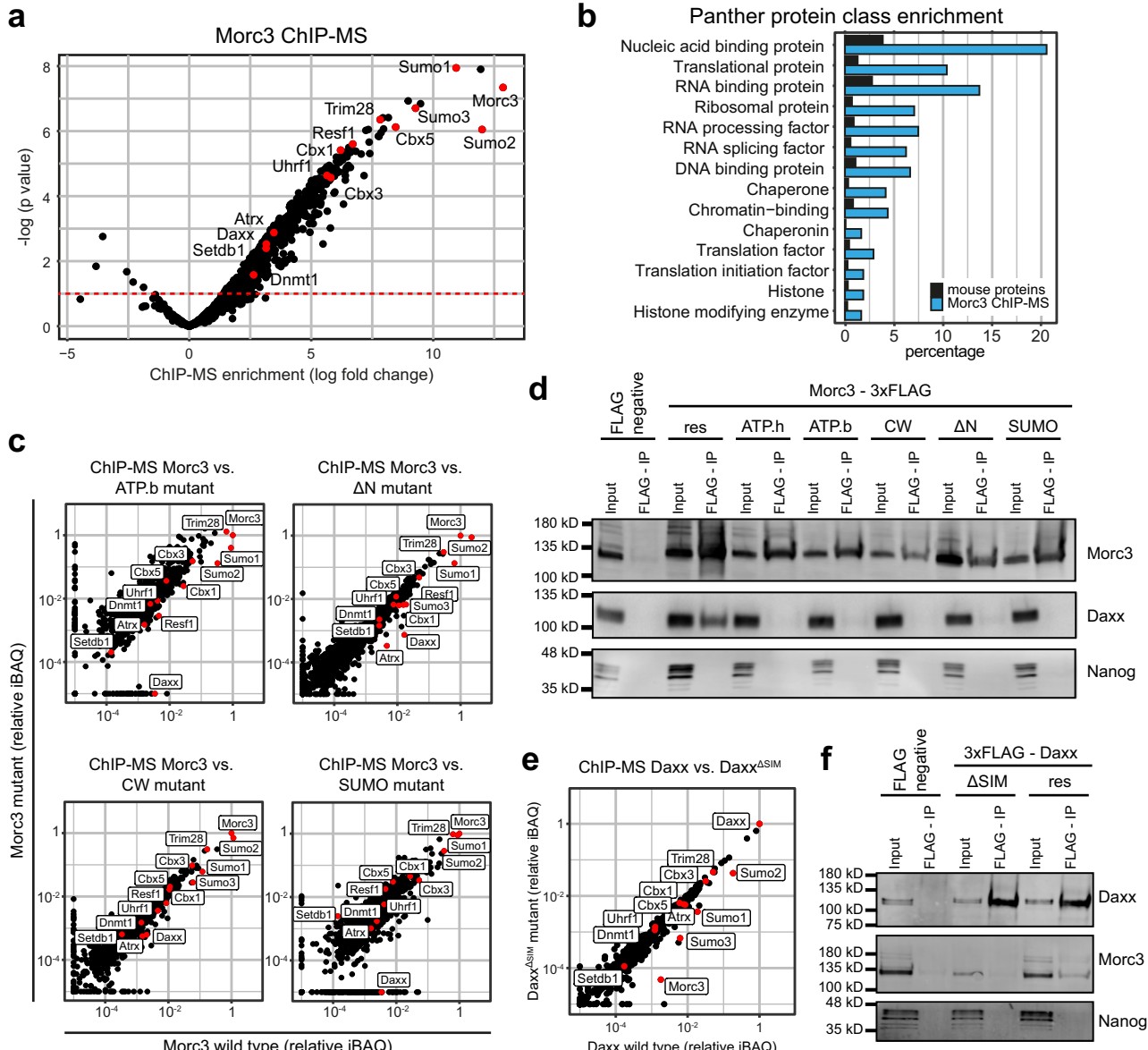

**Fig. 6 Morc3-ATPase cycle and SUMOylation are needed for its interaction with Daxx. a** ChIP-MS identification of Morc3-associated proteins. Morc3 ChIP-MS experiments ($n = 3$) were performed with Morc3-3xFLAG knock-in ES cells and wild-type ES cells using FLAG antibody. Dot plot shows log fold change enrichment vs. $-\log(p$ value) of proteins in Morc3-3xFLAG vs. wild type (background IP) cells. Positions of labeled proteins are indicated by red dots. **b** Panther protein class enrichment analysis of proteins which are significantly enriched in the Morc3 ChIP-MS. Bar graph shows the percentage of total mouse proteins (black) or Morc3 ChIP-MS proteins (blue) in significantly enriched Panther protein classes (Fisher's exact test, $p$ value < 0.05). **c** ChIP-MS analysis of mutant Morc3 proteins. Dot plots show relative iBAQ values of proteins quantified in wild type vs. mutant Morc3 ChIP-MS data. Positions of ERV regulators are indicated. **d** Morc3 interaction with Daxx requires the ATPase cycle and SUMOylation. 3xFLAG tagged Morc3 wild type and mutant proteins were immunoprecipitated from nuclear extract of rescue cell lines using anti-FLAG magnetic beads. Wild-type cells without FLAG epitope served as negative control. Western blot analysis of Morc3, Daxx, and Nanog (negative control) revealed interaction of Morc3 wild type protein with Daxx. Mutant Morc3 proteins fail to co-immunoprecipitate Daxx. Uncropped blots in Source data. **e** ChIP-MS analysis of Daxx wild type vs. Daxx$^{\Delta SIM}$ mutant proteins. Dot plot shows relative iBAQ values of proteins quantified in Daxx rescue cell lines expressing wild-type Daxx or Daxx$^{\Delta SIM}$ mutant protein. Positions of ERV regulators are indicated by red dots. **f** Daxx requires the C-terminal SIM domain for Morc3 interaction. 3xFLAG tagged Daxx wild type and Daxx$^{\Delta SIM}$ proteins were immunoprecipitated from nuclear extract of rescue cell lines using anti-FLAG magnetic beads. Wild-type cells without FLAG epitope served as negative control. Western blot analysis of Daxx, Morc3, and Nanog (negative control) revealed interaction of Daxx wild type protein with Morc3. Mutant Daxx$^{\Delta SIM}$ failed to co-immunoprecipitate Morc3. Uncropped blots in Source data.

LightCycler480 software (Roche) using the 2nd derivative max function. The library preparation was performed in two steps of PCR reactions: For the first PCR, the amount of input genomic DNA (gDNA) for each sample was calculated to achieve 100x coverage over the GECKO library, which resulted in an input of at least 40 μg DNA per sample (assuming 6.6 μg of gDNA for 1 million cells). For the control sample, 46 separate 60 μl PCR reactions were performed with 1 μg gDNA in each reaction using Q5® High-Fidelity DNA Polymerase (New England Biolabs). For the zeocin-treated sample, the total amount of genomic DNA harvested was

used. Oligonucleotides used to amplify lenti CRISPR sgRNAs for the first PCR were GS3367 and GS3368 (Supplementary Table S7). The first PCRs were pooled and 3 μl were used as a template for the second PCR. The second PCR served to attach Illumina adapters and to barcode samples and was done in a 60 μl reaction volume divided to two times 30 μl. Oligonucleotides used as primers for the second PCR include both a variable-length sequence to increase library complexity and an 8 bp barcode for multiplexing of different biological samples Amplification was carried out with 18 cycles for the first PCR and 30 cycles for the second PCR. Primers for

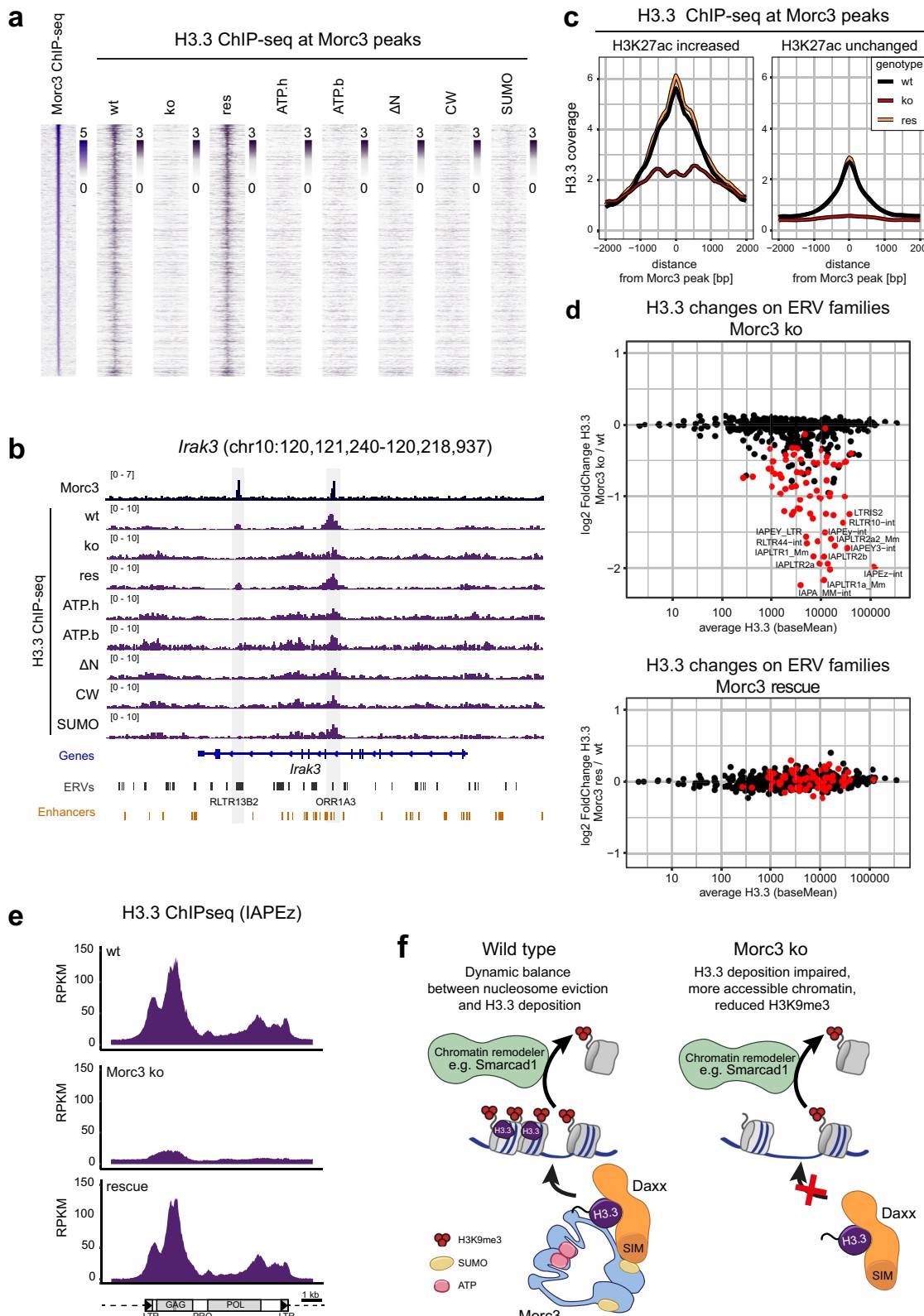

**a** H3.3 ChIP-seq at Morc3 peaks

Morc3 ChIP-seq | wt | ko | res | ATP.h | ATP.b | ΔN | CW | SUMO

**b** *Irak3* (chr10:120,121,240-120,218,937)

Morc3
H3.3 ChIP-seq: wt, ko, res, ATP.h, ATP.b, ΔN, CW, SUMO
Genes — *Irak3*
ERVs — RLTR13B2, ORR1A3
Enhancers

**c** H3.3 ChIP-seq at Morc3 peaks

H3K27ac increased | H3K27ac unchanged

genotype: wt, ko, res

distance from Morc3 peak [bp]

**d** H3.3 changes on ERV families
Morc3 ko

LTRIS2, RLTR10-int, IAPEY_LTR, IAPEy-int, RLTR44-int, IAPLTR2a2_Mm, IAPEY3-int, IAPLTR1_Mm, IAPLTR2b, IAPLTR2a, IAPEz-int, IAPA_MM-int, IAPLTR1a_Mm

average H3.3 (baseMean)

H3.3 changes on ERV families
Morc3 rescue

average H3.3 (baseMean)

**e** H3.3 ChIPseq (IAPEz)

wt, Morc3 ko, rescue

LTR, GAG, PRO, POL, LTR — 1 kb

**f**

Wild type
Dynamic balance between nucleosome eviction and H3.3 deposition

Morc3 ko
H3.3 deposition impaired, more accessible chromatin, reduced H3K9me3

Chromatin remodeler e.g. Smarcad1

Daxx, SIM, Morc3

H3K9me3, SUMO, ATP

the second PCR for zeocin-treated samples were GS3369 and GS3371 (Index9) and for the control samples GS3369 and GS3370 (Index11) (Supplementary Table S7). The resulting amplicons of the second PCR were mixed, purified using AMPure XP beads (Beckman Coulter) in a volume ratio of 1:1 according to manufacturer's instructions. Before sequencing, the purified PCR product was quality controlled by Qubit fluorometer (Invitrogen) and Bioanalyzer (Agilent Technologies) measurement according to manufacturer's instructions. Sequencing was carried out by

LAFUGA sequencing facility at the Gene Center of the LMU Munich using the Illumina HiSeq 1500 and a 50 bp single-end run.

**SHIN silencing assays**

*Initiation assay.* SHIN Reporter constructs (#940 EGFP only and #1074 EGFP SHIN) were stably integrated into cells by lentiviral transduction and the

**Fig. 7 Morc3 is important for Daxx-mediated histone H3.3 incorporation. a** Histone H3.3 is enriched on Morc3 peaks. Read-density heat map showing the normalized coverage of Morc3 and histone H3.3 in wild type, Morc3 ko, and rescue cell lines on Morc3 peaks. The Morc3 ko and all Morc3 mutant rescue cell lines lose histone H3.3 from Morc3 peak regions. **b** Genome browser view of Morc3-dependent histone H3.3 changes on an example target gene (*Irak3*). Positions of two Morc3 peaks are indicated by gray boxes. Both Morc3 binding sites display loss of histone H3.3. Histone H3.3 can be re-established in Morc3 wild type but not mutant rescue cell lines. **c** Density plot showing the average occupancy of histone H3.3 on Morc3 peaks with increased or unchanged H3K27ac in Morc3 ko ES cells. Morc3 peaks with increased H3K27ac signal maintain significant levels of this histone variant while Morc3 peaks with unchanged H3K27ac largely lose histone H3.3. **d** Dot plot showing average histone H3.3 coverage vs. log2-fold change on ERV families in wild type vs. Morc3 knock-out or Morc3 rescue ES cells. Colored dots indicate ERVs significantly enriched with Morc3 (see Fig. 2c). **e** Cumulative coverage plot of histone H3.3 ChIP-seq on IAPEz elements. In wild-type cells, prominent enrichment is over the 5′UTR and the GAG region. The position of the SHIN sequence is indicated as dark gray bar. Morc3 ko cells show a strong reduction of histone H3.3 coverage which is completely reverted in Morc3 rescue ES cells. **f** Model for the role of Morc3 in Daxx-mediated histone H3.3 deposition. In wild-type ES cells, Smarcad1 can evict nucleosomes from ERV chromatin[24]. Nucleosome reassembly requires Daxx-mediated H3.3 incorporation. Our data show that Morc3 interacts with Daxx through the SUMO interaction motif. An efficient Morc3-ATPase cycle and SUMOylation are important for Daxx interaction and assembly of histone H3.3 into ERV chromatin. In Morc3 ko ES cells, Daxx is inactive on ERV chromatin and evicted nucleosomes cannot be replaced. This results in accessible ERV chromatin and reduced H3K9me3.

percentage of EGFP$^+$ cells was measured by FACS after 2–4 days. Lentiviral particles were generated using standard protocols and virus titers were determined by titration in T37 cells. Mouse ES cells were transduced on gelatinized multi-well dishes using spinoculation at a low multiplicity of infection to ensure a linear relationship between virus titer and transduction rate. The ratio of the percentage of EGFP$^+$ cells generated by the reporter relative to the percentage of EGFP$^+$ cells generated by a control EGFP vector of the same virus titer was used to quantify reporter silencing (relative %EGFP$^+$ cells). Sequences of sgRNA oligonucleotides are summarized in Supplementary Table S2.

*Maintenance assay.* Cells containing the SHIN reporter (based on T90 cells) were incubated with 0.1 µg/ml doxycycline for 2 or 4 days and expression of the reporter locus was measured as the percentage of EGFP$^+$ cells. To test the effect of potential ERV silencing mutants, T90 cells were transduced with lentiviral sgRNA constructs. Two days after transduction, cells were selected for sgRNA expression with 1 µg/ml puromycin and SHIN reporter was induced with doxycycline.

**Morc3-FLAG knock-in.** The Morc3-3xFLAG knock-in cell line K14-E8 was generated by CRISPR-Cas9 mediated double-strand break induction close to the Morc3 STOP codon followed by homologous recombination providing a template of the same genomic region (#1529, Plasmids are summarized in Supplementary Table S1), including the 3xFLAG tag before the STOP and exchanging the sequence of the PAM from TGG to TAG so that the repaired sequence cannot be targeted (Supplementary Fig. S1). CRISPR and homology plasmids were transfected via jetPRIME (Polyplus-transfection) and single clones were analyzed for Morc3-3xFLAG expression by western blot. All cell lines used and generated are listed in Supplementary Table 3.

**CRISPR-Cas9-mediated knock-out.** Stable knock-out cell lines were generated by small-guide RNA (sgRNA) mediated Cas9 DNA cleavage using the pX330 plasmid (Addgene plasmid #42230). DNA oligonucleotides were hybridized and ligated into the BbsI digested pX330 to introduce the sgRNA sequence into the vector. Mouse ES cells were co-transfected with the pX330 plasmid harboring the sgRNA and a plasmid encoding a puromycin resistance gene (pLFIP) using jetPRIME (Polyplus-transfection) (#1500/#1501, Supplementary Table S1). After 2 days, transfected cells were selected by addition of 2 µg/ml puromycin to the medium. Puromycin selection was removed after 1 day, and individual cell clones were isolated after 4–6 days. Clonal cell lines were analyzed by western blotting or by PCR and Sanger sequencing.

**Rescue cell lines.** For the generation of Morc3 rescue cell lines, KO27-2 cells were transfected with the PiggyBac (PB) Transposon Vector System via jetPRIME (Polyplus-transfection). The system consists of two plasmids, one encoding the transposase (#1704) and a second one containing the respective rescue construct (Supplementary Table S1). After 2 days, transfected cells were selected by addition of 2 µg/ml puromycin to the medium. Individual cell clones were isolated while puromycin selection was maintained for 7–10 days.

For Daxx rescue cell lines, the Daxx KO cell line KO2-3 was transduced with lentivirus carrying the plasmids #1272 or #1273, respectively (Supplementary Table S1).

**RT-qPCR.** Total RNA was extracted using Trizol and the RNA Clean & Concentrator -25 Kit (Zymo Research, #R1017) including on-column DNAse digestion (Qiagen, #79254) according to manufacturer's instructions. For cDNA synthesis, 1 µg of RNA was used as input. The reaction was carried out using SuperScript III reverse transcriptase (Invitrogen, #18080044), Random Primer 6 (NEB, #S1230S), RNasin Ribonuclease inhibitor (Promega, #N2515). First, total RNA, random

hexamer primers, and RNase-free water were mixed and incubated for 10 min at 70 °C, followed by 1 min incubation on ice. Next, SuperScript buffer, dNTPs, DTT, rRNasin, and SuperScript III reverse transcriptase were added and incubated at 25 °C for 8 min, followed by incubation at 50 °C for 50 min. Reaction was stopped by heat inactivation at 70 °C for 15 min. qPCR was carried out with the Fast SYBR® Green Master Mix (Applied Biosystems, 4385612) in a LightCycler480™ (Roche) according to the Fast SYBR Green Master Mix-protocol. Oligonucleotides used as primers for RT-qPCR are summarized in Supplementary Table S5. Every PCR-reaction was performed in a total volume of 10 µl in triplicates in a 384-well plate (Sarstedt). Two independent control genes (*Actin* and *Hprt*) were used as reference genes for RT-qPCR experiments and geometric mean of reference Ct values was used as normalization. Ct values were generated by the LightCycler480 software (Roche) using the 2nd derivative max function and fold changes were calculated using the $2^{-\Delta\Delta C}$ method.

**ChIP-seq.** For the standard ChIP protocol, 25 million ES cells were cross-linked in 7–8 ml pre-tempered ES medium containing 1% formaldehyde (Pierce #28906) for 10 min at 22 °C. Fixation was stopped by addition of 0.125 M final concentration of glycine, followed by two washing steps with PBS containing 10% Serum. Fixed cells were resuspended in 10 ml ice-cold buffer LB1 (50 mM Hepes-KOH pH 7.5, 140 mM NaCl, 1 mM EDTA, 10% Glycerol, 0.50% NP-40, 0.25% Triton X-100, 1 × Roche cOmplete Mini, EDTA-free Protease Inhibitor Cocktail (Roche, #04693159001), rocked at 4 °C for 10 min and after centrifugation resuspended in 10 ml ice-cold buffer LB2 (10 mM Tris-HCl pH 8.0, 200 mM NaCl, 1 mM EDTA, 0.5 mM EGTA, 1 × Roche cOmplete protease inhibitors), rocked at 4 °C for 5 min and centrifuged again. The pelleted nuclei were resuspended in 1 ml ice-cold shearing buffer (10 mM Tris-HCl pH 8.0, 100 mM NaCl, 1 mM EDTA, 0.5 mM EGTA, 0.1% Na-Deoxycholate, 0.1% SDS, 1 × Roche cOmplete protease inhibitors). Buffer compositions for LB1, LB2, and shearing buffer are based on ref. [49]. Exactly, 1 ml nuclei suspension was transferred to a 1 ml milliTUBE with AFA fiber (Covaris, #520130) and sonicated using a Covaris S220 device for a time of 15–20 min and the following settings: Peak power 140 W, Duty factor 20%, Cycles per burst 200, Temperature 4 °C. The sheared samples were added to a 1.5 ml tube with 110 µl of 10% Triton X-100 (final concentration 1%) and centrifuged at 18407 × g for 15 min at 4 °C. The soluble chromatin containing supernatant was divided in 110 µl aliquots (equivalent to 2–2.5 million cells) and for IP diluted with 890 µl complete Buffer A (10 mM Tris-HCl pH 7.5, 1 mM EDTA, 0.5 mM EGTA, 1% Triton X-100, 0.1% SDS, 0.1% Na-Deoxycholate, 140 mM NaCl, 1 x Roche cOmplete protease inhibitors). Per IP 30 µl of Protein G Dynabeads (Invitrogen, #10004D) were used. For FLAG ChIP-seq experiments 1–2 µl of FLAG-M2 antibody (Sigma-Aldrich, #F3165) were used per IP and 4–5 IPs (equivalent to 10–12 million cells) were pooled to obtain enough material for library preparation. For H3K9me3 ChIP-seq experiments, 1 µl of H3K9me3 antibody (Active Motif, #39161) was used per IP. For H3K27ac ChIP-seq experiments, 2 µg of H3K27ac antibody (Diagenode, pAB-174-050) were used per IP. Beads were incubated with the corresponding antibody in complete Buffer A for 1.5 h prior to IP, followed by two washes with Buffer A (without Roche cOmplete protease inhibitors). Diluted Chromatin was added to the prebound beads and incubated for 4 h on a rotating wheel with 30 rpm at 4 °C. After IP beads were washed five times with Buffer A and 1 time with Buffer C (10 mM Tris-HCl pH 8.0, 10 mM EDTA) beads were resuspended in 100 µl elution buffer (10 mM Tris-HCl pH 8.0, 300 mM NaCl, 5 mM EDTA, 0.5% SDS). RNA was degraded with 2 µl RNase A (10 mg/ml) for 30 min at 37 °C. Proteins were digested with 2 µl Proteinase K (10 mg/ml) at 55 °C for 1 h and cross-link reversal of immunoprecipitated DNA was carried out overnight at 65 °C. DNA was purified using the Agencourt AMPure XP beads (Beckman Coulter, #A63882).

For H3K9me3 and H3K4me3 ChIP-seq in wild type and Morc3 ko cells, a protocol was adapted from ref. [50]. Briefly, 2 million cells were lysed in 100 µl Buffer

B (50 mM Tris-HCl, pH 8.0, 10 mM EDTA, 1% SDS, 1 x Roche cOmplete protease inhibitors) and sonicated in a microtube (Covaris; #520045) using the Covaris S220 device with the settings: Peak power 105 W, duty factor 2%, cycles per burst 200, Temperature 4 °C. The supernatant after centrifugation was diluted with 900 μl complete Buffer A and 150 μl chromatin (corresponding to ~300,000 cells) were used per IP with 10 μl Protein G Dynabeads and 1 μl of H3K9me3 antibody (Active Motif, #39161) or H3K4me3 antibody (Diagenode, CS-003-100) as described above.

Library preparation was performed using the Ultra II DNA Library prep kit for Illumina (NEB, #E7645S) according to manufacturer instructions. Sequencing was performed by LAFUGA on an Illumina Hiseq 1500 using 50 bp paired-end runs for FLAG, H3K9me3 and H3K4me3 ChIPs and a 50 bp single-end run for H3K27ac ChIP.

**Histone H3.3 ChIP-seq.** Histone H3.3 ChIP was performed based on Navarro et al.[24], with slight modifications. In detail, 10 million cells were cross-linked in 3 ml pre-tempered ES medium containing 1% formaldehyde (Pierce #28906) for 10 min at 22 °C. Fixation was stopped by addition of 0.125 M final concentration of glycine, followed by two washing steps with PBS containing 10% Serum. Fixed cells were resuspended in 400 μl of Sonication Buffer 1 (50 mM Tris-HCl (pH 8.0), 0.5% SDS, 1 x EDTA-free Protease Inhibitor Cocktail (Roche, #04693159001)) and sheared with the Bioruptor Pico (Diagenode) for 20 cycles (30 s on/30 s off) at 4 °C. To every 100 μl of sheared chromatin 500 μl lysis buffer (10 mM Tris-HCl (pH 8), 100 mM NaCl, 1% Triton X-100, 1 mM EDTA, 0.5 mM EGTA, 0.1% sodiumdeoxycholate, 0.5% N-laurolsarcodine, 1 × Roche cOmplete protease inhibitors) were added and run full speed at 4 °C for 10 min on a tabletop centrifuge. Each IP used the supernatant corresponding to 3 million cells and 30 μl of Protein A Dynabeads (Invitrogen, #10002D) prebound to 5 μl of H3.3 antibody (Millipore, #09-838) for 1 h at RT followed by two washes with lysis buffer. Chromatin was added to the beads and incubated rotating overnight at 4 °C. The beads were washed twice each time, for 5 min with 1 ml RIPA (10 mM Tris-HCl (pH 8.0), 1% Triton X-100, 0.1% sodium deoxycholate, 0.1% SDS, 1 mM EDTA, 140 mM NaCl), 1 ml RIPA high salt (10 mM Tris-HCl (pH 8.0), 1% Triton X-100, 0.1% sodium deoxycholate, 0.1% SDS, 1 mM EDTA, 360 mM NaCl), 1 ml LiCl Buffer (10 mM Tris-HCl (pH 8.0), 250 mM LiCl, 0.5% NP-40, 0.5% deoxycholate, 1 mM EDTA) and a quick wash with 1 ml Buffer C (10 mM Tris-HCl pH 8.0, 10 mM EDTA). Elution and library preparation was performed as in the ChIP-seq description above with the 50 bp paired-end sequencing mode. All NGS files are listed in Supplementary Table 9.

**ATAC-seq.** The OmniATAC[51] transposition reaction was performed with 50 thousand ES cells using the Tagment DNA TDE1 Enzyme (Illumina, #20034197). DNA was purified using the PCR clean-up MinElute kit (Qiagen, #28006). The transposed DNA was subsequently amplified in 50 μl reactions with custom primers[52] listed in Supplementary Table 8. Libraries were purified and size selected for fragments <600 bp using the Agencourt AMPure XP beads (Beckman Coulter, #A63882). Sequencing was performed by LAFUGA on the Illumina Hiseq 1500 with 50 bp single-end reads.

**RNA-seq.** For samples GS271-GS276, total RNA was extracted using RNeasy Mini Kit (Qiagen, #74106) including on-column DNAse digestion (Qiagen, #79254). Ribosomal RNA was depleted using RNA Ribo-Zero rRNA Removal Kit (Illumina, #MRZH11124). For samples GS947-GS950, total RNA was extracted using Trizol and the RNA Clean & Concentrator -25 Kit (Zymo Research, #R1017) including on-column DNAse digestion (Qiagen, #79254). Due to the discontinued Ribo-Zero kit, ribosomal RNA was depleted with the NEBNext rRNA Depletion Kit (NEB, #E6350). Libraries were prepared with the NEBNext Ultra Directional RNA Library Prep Kit for Illumina (NEB, #E7420S). Sequencing was performed by LAFUGA on an Illumina Hiseq 1500 using 50 bp paired-end runs.

**ChIP-MS.** Immunoprecipitation of bait proteins (Daxx or Morc3) was performed according to the "Rapid immunoprecipitation mass spectrometry of endogenous protein (RIME) for analysis of chromatin complexes" protocol[53]. Cells were harvested and fixed at 22 °C for 10 min in ES medium with 1% Formaldehyde (Pierce #28906) with a density of 4 million cells/ml. The fixation was stopped by 5 min incubation with glycine at a final concentration of 0.125 M. The cell pellet was washed twice with ice-cold PBS containing 10% serum and once with ice-cold PBS without serum. Cell pellets of 60 million cells were flash-frozen and stored at −80 °C for later use. All following steps were performed at 4 °C or on ice and centrifugation was performed at 2000 × g for 5 min if not specified. For each immunoprecipitation experiment, 60 million cells and 60 μl of Protein G Dynabeads (Invitrogen, #10004D) coupled with 6 μl of the FLAG-M2 antibody (Sigma-Aldrich, #F3165) were used. Beads were washed twice with LB3 (10 mM Tris-HCl pH 8.0, 100 mM NaCl, 1 mM EDTA, 0.5 mM EGTA, 0.1% Na-Deoxycholate, 0.5% N-Lauroylsarcosine) and resuspended in 100 μl complete LB3 (LB3 containing 1 × Roche cOmplete Mini, EDTA-free Protease Inhibitor Cocktail (Roche, #04693159001)). The antibody was prebound to beads at 35 rpm on a rotating wheel for 2–4 h and the beads were washed 4 × 1 ml with LB3.

Cells were lysed by resuspending in complete LB1 (50 mM Hepes-KOH, pH 7.5, 140 mM NaCl, 1 mM EDTA, 10% glycerol, 0.5% NP-40, 0.25% Triton X-100, 1 × Roche cOmplete Mini, EDTA-free Protease Inhibitor Cocktail (Roche, #04693159001))

and incubation on a rotating wheel at 35 rpm for 10 min followed by centrifugation. To obtain nuclei, the lysed cells were resuspended in LB2 (10 mM Tris-HCl, pH 8.0, 200 mM NaCl, 1 mM EDTA, 0.5 mM EGTA, Roche cOmplete protease inhibitors) and incubated on a rotating wheel at 35 rpm for 5 min followed by centrifugation. The nuclei were resuspended in complete LB3 (complete LB3, 1x PhosStop EASYpack (Roche, #4906837001), 20 mM N-Ethylmaleimide (NEM) (Thermo Fisher, #23030)) at a maximum density of 20 million cells per 300 μl. Chromatin was sheared with the Bioruptor pico (Diagenode) for 10–12 cycles (30 s on/30 s off) at 4 °C. After shearing 1/10 volume of Triton X-100 was added followed by centrifugation at 20,000 × g for 10 min. The supernatant was added to the beads for chromatin-immunoprecipitation and incubated for 4 h at 35 rpm on a rotating wheel. The beads with the bound chromatin were washed with 4 × 1 ml of RIPA buffer (50 mM Hepes-KOH pH 7.6, 500 mM LiCl, 1 mM EDTA, 1% NP-40, 0.7% Na-Deoxycholate) for 5 min at 20 rpm on a rotating wheel followed by the same washing process with 2 × 1 ml freshly prepared ice-cold 100 mM AMBIC (ammonium hydrogen carbonate) solution. On the last AMBIC wash the beads were transferred to a new tube. The beads were stored dry at −80 °C until MS analysis.

The sonicated chromatin was size checked with a 2100 Bioanalyzer (Agilent) using the DNA high sensitivity kit (Agilent, #5067-4626) or the DNA 7500 kit (Agilent, #5067-1506) following manufactures protocol. Shortly, Elution buffer (0.5% SDS, 300 mM NaCl, 5 mM EDTA, 10 mM Tris-HCl) was added 1:10 to the sonicated chromatin and incubated at 65 °C, shaking at 800 rpm overnight for de-cross-linking. The de-cross-linked chromatin was treated with 0.2 μg/ml of RNAse A and incubated at 37 °C, shaking at 700 rpm for 1 h. Next, the sample was treated with 0.2 μg/ml Proteinase K and incubated at 55 °C, shaking at 700 rpm for 2 h. The DNA was purified with the MinElute PCR purification kit (Qiagen, #28006) and concentration was measured with the Qubit dsDNA high sensitivity kit (Thermo Fisher, #Q32851) before loading the samples to the DNA HS/DNA 7500 chip.

**Liquid chromatography–mass spectrometry (LC–MS).** Beads were washed three times with 50 mM $NH_4HCO_3$ and incubated with 0.5 μg/μl Lys-C and 20 U benzonase in 6 M urea, 50 mM $NH_4HCO_3$ pH 7.5 for 90 min at 28 °C, washed with 50 mM $NH_4HCO_3$, and the combined supernatants were digested overnight with 0.2 μg/μl trypsin in presence of 10 mM DTT. Digested peptides were alkylated with 30 mM IAA and desalted prior to LC–MS analysis.

For LC–MS/MS purposes, desalted peptides were injected in an Ultimate 3000 RSLCnano system (Thermo), separated in either a 15 cm analytical column (75 μm ID with ReproSil-Pur C18-AQ 2.4 μm from Dr. Maisch) with a 50 min gradient from 4 to 40% acetonitrile in 0.1% formic acid or in a 25 cm analytical column (75 μm ID, 1.6 μm C18, Aurora-IonOpticks) with a 50 min gradient from 2 to 35% acetonitrile in 0.1% formic acid. The effluent from the HPLC was directly electrosprayed into a Qexactive HF (Thermo) operated in data-dependent mode to automatically switch between full-scan MS and MS/MS acquisition. Survey full-scan MS spectra (from m/z 375 to 1600) were acquired with resolution $R = 60,000$ at m/z 400 (AGC target of $3 × 10^6$). The 10 most intense peptide ions with charge states between 2 and 5 were sequentially isolated to a target value of $1 × 10^5$ and fragmented at 27% normalized collision energy. Typical mass spectrometric conditions were spray voltage, 1.5 kV; no sheath and auxiliary gas flow; heated capillary temperature, 250 °C; ion selection threshold, 33,000 counts.

**Nuclear extraction.** Nuclei for IP and western blot analysis were isolated via ficoll gradient centrifugation. For each isolation 20 million cells were harvested and overlayed on a ficoll gradient consisting of 20% Ficoll (Lymphocyte Separation Medium1077, promo cell C-44010) and 80% NI-Stock (100 mM Tris/HCl pH 7.4, 10 mM $MgCl_2$, 10 mM $CaCl_2$l, 2% NP-40, 1.6% Triton X-100) with 0.1% DMSO. Centrifugation speed was stepwise increased at 4 °C from 37 to 150 × g in 8 min (30 s at 37 × g, 30 s at 58 × g, 30 s at 84 × g, 30 s at 114 × g, 6 min at 150 × g). The supernatant was carefully removed, and nuclei washed once in cold PBS.

**Co-IP for western blot.** Nuclei were isolated via ficoll gradient centrifugation and resuspended in complete 150 mM NaCl IP Buffer (50 mM HEPES ph 7.5, 150 mM NaCl, 0.05% NP-40, 20% glycerol and freshly added 1 × Roche complete, 1 × PhosphoSTOP, 25 mM NEM, Benzonase (Merck, #1016540001) (1 μl for each 400 μl of buffer) and 2.5 mM $MgCl_2$). Tubes were incubated 3 min at 37 °C for benzonase digestion, then salt concentration was adjusted to 300 mM by adding 5 M NaCl. After 30 min incubation on ice samples were centrifuged 30 min at 4 °C at maximum speed and the supernatant was used as input for IP. Since the bait proteins were FLAG-tagged, FLAG-M2 beads (Sigma, M8823-1ML) were used (80 μl beads for 80 million cells). IP was performed for 4 h followed by three washing steps, 5 min each with 1 ml 300 mM NaCl IP buffer (50 mM HEPES ph 7.5, 300 mM NaCl, 0.05% NP-40, 20% glycerol) on a rotating wheel at 4 °C. For elution of immunoprecipitated proteins, beads were first boiled in Lämmli buffer without β-mercaptoethanol (for Daxx elution) and then boiled in Lämmli buffer containing β-mercaptoethanol (for Morc3 elution). Morc3 western blot signals were detected by multiplexed fluorescent immunoblotting (LI-COR Biosciences). Development of Daxx western blots after co-IP was performed with the SuperSignal™ Western Blot Substrate Atto (Thermo Scientific, A45918).

**Immunofluorescence analysis.** Cells were seeded in 24-well plates containing 12 mm poly-L-lysine coated coverslips (50 thousand cells/well). Cells were fixed with

formaldehyde (500 μl/well, 3.7% formaldehyde in PBS for 10 min at RT). Coverslips were washed twice for 5 min in 1 ml PBS. Permeabilization was performed for 5 min in 500 μl/well in permeabilization solution (10 mM sodium citrate tribasic dihydrate, 200 μl Triton X-100). Two washing steps with 1 ml PBS followed by two washing steps with 1 ml washing solution I (2 l $H_2O$, 220 ml 10 x PBS, 2.2 ml Tween-20, 5.5 g BSA) were performed before samples were blocked for 30 min in 300 ml blocking solution I (50 ml washing solution I, 1.2 g BSA). Primary antibody (anti-Morc3, Rockland, 100-401-N96S) was diluted in 200 μl/well blocking solution I and incubated over night at 4 °C in the dark. Cells were then washed three times with washing solution I. Secondary antibody was diluted in 200 μl/well blocking solution II (0.5 ml blocking solution I, 0.5 ml serum (goat or donkey)) and incubated for 1 h at RT in the dark. Next, coverslips were washed three times 10 min with washing solution II (500 ml PBS, 500 μl Tween-20), followed by embedding coverslips with Vectashield containing DAPI (Vector Laboratories, Axxora - H-1200) and sealed with nail polish. Samples were stored at 4 °C in the dark until pictures were taken at inverted confocal microscope Leica SP5 (×64) and analyzed with ImageJ.

**Statistics and reproducibility**. Results shown in Fig. 1e, g are combinations of several experiments and the replicate number per sgRNA varies. Specifically in Fig. 1e $n = 4$ for sgDaxx, sgDnmt1, sgSetdb1 and Trim28; $n = 3$ for sgAtrx, sgMorc3, sgNeg, SgTrim24 and sgUhrf1, $n = 2$ for all remaining candidates. In Fig. 1g: $n = 12$ for sgNeg, $n = 11$ for sgSetdb1, $n = 10$ for sgDaxx, $n = 7$ for sgDnmt1 and sgUhrf1, $n = 6$ for no sgRNA control, sgMorc2a and sgTrim28, $n = 4$ for sgAtf7ip and sgAtrx, $n = 3$ for Morc3 and sgZfp606, $n = 2$ for sgDek and sg HP1, $n = 1$ for all remaining candidates.

All western blots are reproduced with $n \geq 2$ biologically independent experiments. All uncropped blots are present in the source data file.

### Bioinformatics analysis

*sgRNA screen data analysis*. Fastq files were trimmed with Trimmomatic version 0.36 and options "CROP:43 HEADCROP:23" to contain only the unique sgRNA sequence. Reads were then mapped with bowtie to all sgRNA sequences of the Gecko V2 library. Cvs files of library sequences were downloaded from addgene (https://www.addgene.org/pooled-library/zhang-mouse-gecko-v2/). CSV files were converted to fasta format using R. Bowtie was used to generate the library index. Reads were then aligned with bowtie 1. The number of aligned reads to each sgRNA sequence was calculated with bedtools command "bedtools genomecov". The numbers of reads for each sgRNA per sample were normalized in R as follows: Normalized reads per sgRNA = (reads per sgRNA/total reads for all sgRNAs in sample) + 1. Hits were identified by conversion of sgRNA enrichment scores into gene rankings by statistical analysis using the SecondBestRank scoring method for RNAi gene enrichment ranking (RIGER) with the RigerJ tool.

*ChIP-seq*. Paired-end ChIP-seq reads were aligned to the mouse genome mm10 using Bowtie2 with default settings. The resulting BAM files were filtered to remove non-paired reads, low mapping quality, non-primary alignment, and PCR duplicates with "samtools view -b -f 2 -F 1280 -q 20". Homer tag directories were generated with "makeTagDirectory". BigWig files were generated from tag directories with "makeBigWig.pl mm10 -webdir. -url. -norm 1e7 -normLength 100 -fragLength 150 -update".

Morc3 peaks were identified using findPeaks with option "-style factor" over Input. The peak files of replicate experiments were merged using homer "mergePeaks" and, peaks common in two replicates were kept for the final peak list. The final Morc3 peak list was annotated using homer "annotatePeaks.pl", resulting in detailed peak annotation including association with repeat elements.

Morc3 association with repeats was determined using RepEnrich2. The fraction counts for each repeat family were normalized and repeat families with significant enrichment (padj < 0.05 & log2FoldChange > 0.5) in Morc3-FLAG ChIP-seq vs. wild type FLAG ChIP-seq (background) were calculated using DeSeq2. Enrichment of H3K9me3 on ERVs in wild type vs. Morc3 knock-out and Morc3 rescue cells was also calculated with RepEnrich2 and DeSeq2.

Heatmaps for ERV silencing factors Trim28, Setdb1, Morc3, and histone H3.3/H3K9me3 were plotted as log-transformed normalized coverage (calculated from Homer tag directories with annotatePeaks.pl).

Cumulative coverage plots on IAPEz elements were performed as described earlier[20]. Cumulative coverage plots on IAPEz of H3K9me3 mutant rescues were normalized based on non-Morc3-related H3K9me3 peaks in the wt sample.

Heatmaps of H3K9me3 and H3K27ac were plotted as log-transformed normalized coverage (calculated from Homer tag directories with annotatePeaks.pl). Heatmaps of H3K9me3 mutant rescues were normalized based on non-Morc3 related H3K9me3 peaks in the wt sample.

*RNA-seq*. Paired-end reads were aligned to the mouse genome version mm10 using STAR with default options "--runThreadN 32 --quantMode TranscriptomeSAM GeneCounts --outSAMtype BAM SortedByCoordinate". Read counts for all genes were normalized using DESeq2. Differentially expressed genes were determined using the DESeq2 results function (adjusted *p*-value < 0.01).

Expression of ERV families was calculated using RepEnrich2. Fraction counts were normalized and differentially expressed repeats were calculated using DeSeq2.

*ATAC-seq*. Single-end ATAC-seq reads were aligned to the mouse genome mm10 using Bowtie with options "-q -n 2 --best --chunkmbs 2000 -p 32 -S -m 1". Duplicated reads were subsequently removed using Picard. Homer tag directories were generated using makeTagDirectory and bigwig files were generated with makeBigWig.pl. ATAC peaks were identified using Homer findPeaks.pl with the option "-style factor" over Input. Peaks from all samples were merged using mergePeaks resulting in a unified Peak set. Raw ATAC coverage counts were calculated with annotatePeaks. Differential ATAC peaks were determined with the DESeq2 result function.

ATAC coverage of ERV families was calculated using RepEnrich2 on BAM files including multi-mapping reads. Fraction counts were normalized, and differentially accessible repeats were calculated using DeSeq2.

*IGV Screenshots*. IGV screenshots were taken from respective bigwig files using IGV 2.10.0.

*ChIP-MS data analysis*. MaxQuant (1.6.14.0) was used to identify proteins and quantify by iBAQ with the following parameters: Database, UP000000589_10090_Mmusculus_2020; MS tol, 10 ppm; MS/MS tol, 20 ppm Da; Peptide FDR, 0.1; Protein FDR, 0.01 Min. peptide Length, 7; Variable modifications, Oxidation (M); Fixed modifications, Carbamidomethyl (C); Peptides for protein quantitation, razor and unique; Min. peptides, 1; Min. ratio count, 2.

The final list of proteins found was filtered and statistically processed in R studio version 3.5.0 using the Linear Models for Microarray Data (LIMMA) R script Version 1.0.1 that is available on GitHub written by Wasim Aftab (https://github.com/wasimaftab/LIMMA-pipeline-proteomics).

For comparative analysis between Morc3 wild type and mutant proteins, iBAQ values for proteins identified with more than 2 peptides were normalized to Morc3 or Daxx, respectively, by dividing by the iBAQ value of the bait. Mean normalized iBAQ values from replicate experiments were plotted. Daxx was not detected in the Morc3 ATP binding and SUMOylation mutant ChIP-MS experiments.

Pathway enrichment analysis of significantly Morc3-associated proteins was performed using Panther.

**Reporting summary**. Further information on research design is available in the Nature Research Reporting Summary linked to this article.

## Data availability
The data that support this study are available from the corresponding author upon reasonable request. ChIP-MS proteomics datasets generated in this study have been deposited to the ProteomeXchange database under the accession code PXD027368. ATAC-seq, ChIP-seq, and RNA-seq datasets generated in this study have been deposited to the GEO database under the accession code GSE159936. The published Trim28 ChIP-seq data used in this study are available in the GEO database under accession code GSM1819199. The published Daxx-ko and H3.3-ko RNA-seq data used in this study are available in the GEO database under accession code GSE102688. Source data are provided with this paper.

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

## Acknowledgements

High throughput sequencing was performed by the Laboratory for Functional Genome Analysis (LAFUGA) of the Ludwig-Maximilian-University, Munich. We acknowledge the Core Facility Flow Cytometry and the Bioinformatics Core Unit at the Biomedical Center, Ludwig-Maximilian-Universität München, for providing equipment, service, and expertise. Mouse GeCKOv2 CRISPR knock-out pooled library was a gift from Feng Zhang (Addgene #1000000052). The NP95 antibody was a gift from Heinrich Leonhardt (Faculty of Biology, Ludwig-Maximilians-University (LMU) Munich, Großhaderner Straße 2, 82152 Martinsried, Germany). Funded by the Deutsche Forschungsgemeinschaft (DFG, German Research Foundation) – Project-ID 213249687 – SFB 1064 TP3 and Projekt-ID 329628492 – SFB 1321 TP13) to G.S.

## Author contributions

Conceptualization: S.G. and G.S. Investigation: S.G., A.V.M., L.K.M., C.V.S., A.R., H.B., G.P.dA., A.S., and I.F. Data curation: S.G., A.V.M., L.K.M., C.V.S., A.R., H.B., G.P.dA., A.S., and I.F. Formal analysis: S.G., A.V.M., A.S., I.F., and G.S. Funding acquisition: G.S. Methodology: S.G. and G.S. Project administration: S.G. and G.S. Resources: A.I. and G.S. Software: S.G., A.S., I.F., and G.S. Supervision: S.G. and G.S. Validation: S.G., A.V.M., L.K.M., C.V.S., A.R., H.B., and G.S. Visualization: S.G., H.B., C.V.S., A.V.M., A.R., and G.S. Writing—original draft: S.G., G.S. Writing—review & editing: S.G., A.V.M., L.K.M., C.V.S., A.R. H.B., G.P.dA., A.S., I.F., A.I., and G.S.

## Funding

## Competing interests

The authors declare no competing interests.
