## [Peer Review File · Nature Communications]

REVIEWER COMMENTS

Reviewer #1 (Remarks to the Author):

In this interesting manuscript by Groh et al. the role of Morc3 in silencing of ERVs in mouse embryonic stem cells is investigated. The manuscript is mostly well written and easy to understand. The topic is of high importance and the study is timely. Overall, I found that this study contained many interesting experiments. Data related to the role of Morc3 as a compactor of chromatin are particularly convincing. The observation of specific H3K9me3 depletion around Morc3 binding sites is intriguing. However, I also found that several aspects of the study, in particular the last part, were rather preliminary and would benefit from additional experiments.

Major comments:

1. The authors interpretation of the results are at several places overstated. For example, in the abstract they claim that Morc3 is “critical for ERV silencing”. A more proper interpretation would be that Morc3 is a regulator of chromatin structure that modestly impacts on ERV silencing. This should be modified throughout the manuscript.
2. The fact that ERVs are only modestly upregulated in Morc3-KO cells makes the biological significance uncertain. Is this an in vitro phenomenon? It would be extremely valuable to investigate ERV-expression in early embryos from Morc3-KO mice.
3. The first two paragraphs of the introduction do not properly represent the current state of the field. In addition, key references are missing (eg. Rowe et al., Nature 2010 & Matsui et al., Nature 2010). These two paragraphs should be modified to better represent the current state of the literature.
4. The use of the Morc3 rescue lines is potentially very valuable. What happens with endogenous ERV expression in this line? Is ERV repression restored?
5. I like the idea of using specific Morc3 mutants (E35A, G101A, W419A) to dissect function, but the data may well be misinterpreted. For example, the blot in Fig S5 shows a faint band for E35A alongside rather featureless tracks, and MORC structural data would suggest that any mutation in

the active site could compromise the fold of the ATPase module. This is important because if the protein is improperly folded, its interactome and chromatin association will change. More controls are needed. The authors seem to have taken the mutants from Mimura et al (2010) but knowledge of the structures and activities of MORCs has moved on substantially since then. The interpretation on the mutants is also a bit confusing. The statement 'E35A is impaired to dissolve the ATPase dimer, and G101A is unable to dimerize' is questionable. To my knowledge this hasn't been shown for Morc3. It implies E35A would be a constitutive dimer and that G101A would be able to bind ATP. However, since binding and dimerization are coupled for the MORCs (Li et al, 2016; Douse et al 2018), these mutants may not be doing what the authors claim.

6. The fact that 489 interacting proteins were found in the mass-spec analysis suggest that there might be a problem with false positives. Validation of some of these interactions (at least between Morc3 and Trim28, Setdb1, Daxx) with an alternative technique, such as a classic antibody-based IP, or with purified components, is essential.

7. ChIP-seq data of H3.3 in Morc3-wt/KO cells would be very valuable in order to interpret the importance of this potential interaction between Morc3/Daxx/H3.3

8. Are the same ERV-integrants activated in H3.3-KO, Daxx-KO and Morc3-KO cells?

9. The discussion is highly speculative regarding the interaction between Morc3 and Daxx. This part of the manuscript should be modified to better represent the presented data.

10. The ChIP-seq data are in conflict with published data (Li et al 2016) showing association with H3K4me3 over promoter regions. In that case, a Morc3 antibody was used, rather than a knock-in strategy. Could the authors try to resolve the apparent discrepancy? There is also convincing structural and biophysical data using purified components from multiple labs that Morc3 binds to unmodified H3 and H3K4me3 but NOT H3K9me3. Indeed, in Li et al, H3K9me3 was a negative control since 'it was anti-correlated with morc3 in the genome'. A manual inspection of the chip-seq on morc3 looks ok on first glance. But inclusion of supp figures with controls (e.g. show the input track, show their other negative controls e.g. the anti-FLAG track in parent cells, to check for non-specific CHIP) would be valuable.

11. The authors should compare the results of their screen to a similar recent screen in mESCs (Bourc'his lab, <https://www.biorxiv.org/content/10.1101/2020.03.24.005488v1>)

12 Is it correct that nk the rescue rna-seq only has 1 replicate? If so, please perform additional

13 The authors should state that the setdb1 chip is from a knock in

14. It is unclear where the trim28 chip-seq comes from

Reviewer #2 (Remarks to the Author):

Groh et al.

This paper reports the finding, through a genome-wide CRISPR KO screen, that the Morc3 gene is important in silencing specific families of the ERVs in ES cells. Morc3 KO cells show reduced silencing, altered chromatin marks. Wild-type Morc3 restored silencing, but ATPase mutants did not. The paper shows that Morc3 interacts with the H3.3 chaparone Daxx, and that this required ATP binding, and Morc3 SUMOylation. Finally, Morc3 KO cells show reduced H3.3 at Morc3 binding sites.

The target locus used for the screen is a short element in the Gag region of the IAP family of ERVs, one previously identified by the lab (as a SHIN, short heterochromatin inducing sequence), which is used to regulate a dox-inducible promoter of reporter gene expression. Hits were validated for both maintenance of silencing and establishment of new silencing, leading to Morc3 (among other hits).

Chip-seq showed that Morc3 was localized to many ERVs, overlapping with sites of Setdb1 and Trim28 binding. Morc3 KO ES cells showed upregulation of many genes, three of which are noted to be near Morc3 binding sites. They also showed upregulation of some ERVs, though not as strongly as Setdb1 KO. There were complex changes in H3K9me3 patterns at Morc3 binding sites. ATAC seq revealed hundreds of sites with increased accessibility in the KO cells.

Proteomic analysis of CHIP samples revealed co-IP of Morc3 with hundreds of proteins, including Daxx, which bound to wt Morc3 but not to an ATP binding site mutant. Daxx is a known heterochromatin silencer, working through a SUMO interaction motif, to mediate H3.3

incorporation. H3.3 ChIP revealed overlap with Morc3 sites, and reduced signal at many sites in Morc3 KO cells.

The model is presented that normally there is ongoing, rapid turnover of nucleosomes in silent heterochromatin, and that Daxx is needed to re-establish H3.3 and maintain silencing, and that this requires Daxx interaction with SUMOylated Morc3. An active ATPase cycle is also proposed to be needed for Morc3 function.

This is a very nice paper with an impressive amount of new information about ERV silencing. Morc1 and Morc2 have been shown to be important for silencing of ERVs in other settings, and here this extended to Morc3; and we also learn of its function through Daxx and H3.3 and H3K9me3. This is close to a perfect paper: the study is well-designed, the experiments are strong, the data are all solid, and the paper is clear and well-written. There is a huge amount of work presented, and there is much new information that will be of interest to the community.

I can find nothing significant to suggest. Perhaps a few comments on the history of the SHIN site would be helpful.

Reviewer #3 (Remarks to the Author):

In this manuscript, Groh et al identify the ATPase Morc3 as a novel protein involved in ERV silencing in mouse embryonic stem cells. Loss of Morc3 leads to loss of H3K9me3 and increased chromatin accessibility at a subset of ERVs. The authors then study the function of three established Morc3 mutants. They find that (1) an ATP hydrolysis mutant cannot bind to chromatin, and (2) an ATP binding mutant cannot bind to DAXX, a protein previously identified by several groups to be involved in ERV silencing. Interaction between Morc3 and DAXX requires the SIM domain of the latter. The model put forth is that Morc3-DAXX binding is required for DAXX recruitment to ERVs, H3.3 deposition, and subsequent chromatinization and heterochromatinization.

The validation of Morc3 as an ERV silencing protein using the reporter plasmid and stable reporter ESCs are clean and convincing. The ChIP-seq data localizing Morc3 to ERVs is also straightforward and in agreement with the screen validation. I have some minor concerns about the integration of various omics data sets in Figures 3 and 4. I have more serious concerns about how well claims from Figures 5 and 6 have been substantiated to support the model put forth in Figure 7. Should the authors provide more evidence in support of this model, I think this work will make an important contribution towards our understanding of heterochromatin formation at ERVs.

Suggested improvements in computational analysis:

1. The authors suggest that Morc3 is suppressing LTRs which may otherwise act as enhancers for nearby genes. Do these regions gain marks of active enhancers such as p300 binding and H3K27ac?
2. The authors identify 64 upregulated genes with Morc3 peaks <100kb. From Fig. 3B, it looks as though the authors have RNA-seq from rescue ESCs. This would allow the authors to provide a more comprehensive assessment of whether Morc3 OE can rescue (1) all global transcription defects observed in the KO, or perhaps (2) a subset of DE based on proximity of a Morc3 binding site to the promoter as delineated in Supplemental Table 5. This would be preferable over the select RT-PCR shown in Fig. 3C.
3. I'd like to see some statistics associated with the H3K9me3 ChIP-seq data in Figure 4A. The hierarchical clustering and heat map that has been performed does not allow this.
4. It's interesting that Morc3-bound sites show differential effects on H3K9me3 when Morc3 is lost. This does not appear related to the level of Morc3 enrichment within a particular group. Is there any other genomic feature that influences how sensitive a particular region is to Morc3 loss?
5. The directionality of the H3K9me3 loss in Morc3 KO cells is quite interesting and I'd be curious to know more about the mechanism behind this.
6. The ATAC-seq data are poorly linked back to Morc3 binding. The authors anecdotally claim that differentially regulated ATAC peaks are generally within proximity of a Morc3 peak, but a few browser tracks are not enough to make this claim and this point must be fully addressed computationally.
 - a. What is the distribution of distance between Morc3 peaks and ATAC peaks?

b. Do all Morc3-bound peaks become more open in the absence of Morc3 and is the difference statistically relevant?

c. Is differential accessibility correlated with decreased H3K9me3?

Overall, while Figure 4D is quite striking, the earlier data in this figure lack appropriate depth of analysis and comparative analysis.

7. The authors have performed ATAC-seq using their Morc3 mutants (Fig. 5E and Supplemental Fig. 6), but they do not present this genomic analysis. Again, browser track snapshots of a few regions are not sufficient when genome-wide analyses can be performed.

Other major points:

8. It's really critical that the authors address head on that the reported ligand for the CW domain is H3K4me3. The intro to this section is difficult to understand because the authors seem not to want to state this. While they do bring up this point in the discussion, I don't feel that it has been sufficiently addressed. Based on Fig. 2A there is very little Morc3 binding at promoters (sites of H3K4me3 enrichment) and the authors state that they do not observe H3K4me3 enrichment in ERV heterochromatin. It's difficult to understand why the CW domain should be required for their SHIN silencing assay or for transcription regulation as shown in Figure 5C and 5D. Morc3 localization seems least affected by the CW mutation, but the authors do not provide a satisfying explanation as to why this region is important for ERV silencing.

9. In general, the proteomics analyses should be validated using other biochemical methods. The model is that Morc3 is a direct binding partner of DAXX mediated by Morc3 SUMOylation and the DAXX SIM domain, but the data at this stage do not fully support this model. If the site of SUMOylation on Morc3 is known, this would be a valuable mutant in this study.

10. The model is that the entire ATP hydrolysis cycle of Morc3 is required for ERV silencing by regulating DAXX-mediated H3.3 deposition. This point needs additional experimental support. Given that its expression levels are low and the mutant does not bind to Morc3 target regions, I assume the ATP hydrolysis mutant acts as a Morc3 KO. However, the authors should more fully characterize the CW and ATP-binding mutants. It's quite curious that the CW mutant maintains all SETDB1-complex associations and is retained on chromatin yet results in SHIN reporter activation. What, then, is the function of the CW domain? Does CW mutant expression affect H3.3 deposition or H3K9me3 levels at ERVs? Further, the ATP-binding mutant binding to chromatin seems attenuated (see Supplemental Fig. 5B, 5C), yet this mutant maintains all SETDB1-complex associations (including ATRX) with the striking exception of DAXX. This result needs orthogonal validation. In addition, the same H3K9me3 and H3.3 genomic profiling should be completed from cells expressing the ATP.b mutant. The authors should also test whether each mutant affects nuclear DAXX levels. Ideally,

DAXX ChIP should be provided as well to determine whether DAXX is lost from chromatin when Morc3 cannot bind ATP.

Minor Points:

1. I'm not sure why the authors state in Line 61 that no screens have been performed using an endogenous ERV sequence as they themselves published such an shRNA screen in their 2015 EMBO Rep paper.
2. Validation of Morc3-3xFlag is lacking (i.e., genotyping, Flag western blot).
3. Authors should provide a table to better document all sequencing data used in this study.
4. There are some issues with the callouts for Figure 4.
5. Lines 207-210 are speculative. ATAC-seq does not measure whether TFs are bound, it measures whether chromatin is accessible. These lines in particular, but there are a number of concluding statements interspersed throughout the results that do not describe the results as presented and are speculative in nature. These would be better suited for the discussion.
6. Supplemental Fig. 5C should be on the same scale. Both ATP binding mutant and ATP hydrolysis mutant look reduced at target sites. They also both look more lowly expressed compared to WT rescue.

REVIEWER COMMENTS

Reviewer #1 (Remarks to the Author):

In this interesting manuscript by Groh et al. the role of Morc3 in silencing of ERVs in mouse embryonic stem cells is investigated. The manuscript is mostly well written and easy to understand. The topic is of high importance and the study is timely. Overall, I found that this study contained many interesting experiments. Data related to the role of Morc3 as a compactor of chromatin are particularly convincing. The observation of specific H3K9me3 depletion around Morc3 binding sites is intriguing. However, I also found that several aspects of the study, in particular the last part, were rather preliminary and would benefit from additional experiments.

Major comments:

1. The authors interpretation of the results are at several places overstated. For example, in the abstract they claim that Morc3 is “critical for ERV silencing”. A more proper interpretation would be that Morc3 is a regulator of chromatin structure that modestly impacts on ERV silencing. This should be modified throughout the manuscript.

We changed the wording according to the reviewer’s suggestion.

2. The fact that ERVs are only modestly upregulated in Morc3-KO cells makes the biological significance uncertain. Is this an in vitro phenomenon? It would be extremely valuable to investigate ERV-expression in early embryos from Morc3-KO mice.

Our study is focused on the mouse ES cell system. In these cells, we detected Morc3-dependent changes in ERV chromatin. However, transcriptional effects cannot be detected on all Morc3 binding sites. The reason is most likely compensation by additional silencing machineries that counteract transcriptional activators. It is, however, important to note that we do detect genomic regions where Morc3 binding is critical for full transcriptional repression. These Morc3 binding sites are present in two distinct genomic regions: (1) ERV integrations with mainly LTRIS2 elements and (2) enhancer regions characterized by low-level H3K27ac. We find Morc3-dependent transcriptional de-repression in the context of both categories (examples shown in IGV screenshots are Ube2l6, Cd200 and Irak3). In summary we think that Morc3 is a contributing factor for silencing, which, in some genomic regions with little compensation by other mechanisms, becomes critical for silencing. It is very likely that these genomic regions are cell type-specific and that in a different cell type, other genomic elements may be targeted by Morc3 or may show transcriptional responses to Morc3 loss. We think this is an important question that we will examine in follow up studies.

3. The first two paragraphs of the introduction do not properly represent the current state of the field. In addition, key references are missing (eg. Rowe et al., Nature 2010 & Matsui et al., Nature 2010). These two paragraphs should be modified to better represent the current state of the literature.

The first paragraph should highlight that ERVs contribute to normal development and can be dysregulated in disease. This is to highlight the importance of ERV regulation, rather than a comprehensive review of the literature in this field. The second paragraph is more towards the transcriptional regulation of ERVs by chromatin. Here we summarize the major players that have

been described in this context, although the mechanism of action is largely unclear. We cite the recent review by Geis and Goff, which is a very nice and comprehensive summary of this field.

As suggested by the reviewer we cite more key references on the Setdb1/Trim28 pathway.

4. The use of the Morc3 rescue lines is potentially very valuable. What happens with endogenous ERV expression in this line? Is ERV repression restored?

We now provide a deeper analysis of transcriptional changes in Morc3 rescue cells (new Supplementary Figure S4). We detect that both, gene and ERV expression, are largely restored in Morc3 rescue cell lines.

5. I like the idea of using specific Morc3 mutants (E35A, G101A, W419A) to dissect function, but the data may well be misinterpreted. For example, the blot in Fig S5 shows a faint band for E35A alongside rather featureless tracks, and MORC structural data would suggest that any mutation in the active site could compromise the fold of the ATPase module. This is important because if the protein is improperly folded, its interactome and chromatin association will change. More controls are needed. The authors seem to have taken the mutants from Mimura et al (2010) but knowledge of the structures and activities of MORCs has moved on substantially since then. The interpretation on the mutants is also a bit confusing. The statement 'E35A is impaired to dissolve the ATPase dimer, and G101A is unable to dimerize' is questionable. To my knowledge this hasn't been shown for Morc3. It implies E35A would be a constitutive dimer and that G101A would be able to bind ATP. However, since binding and dimerization are coupled for the MORCs (Li et al, 2016; Douse et al 2018), these mutants may not be doing what the authors claim.

In the revised manuscript we added the analysis of two additional Morc3 mutants. The Morc3 SUMOylation mutant (also from Mimura 2010) cannot be SUMOylated and fails to interact with Daxx. This supports SUMO-dependent interactions between Morc3 and Daxx. The ΔN mutant is from Zhang 2019 and partially characterized as non-dimerizing and having reduced ATPase activity. The "old" mutants from Mimura 2010 (ATP hydrolysis and ATP binding) are not structurally characterized. However, they are modeled after mutations in Hsp90, which did not impair overall structure of the protein. Nevertheless, we agree that precise interpretations are difficult without a comprehensive structural analysis in combination with our functional data. Therefore, we changed our line of argumentation. Having different mutants that impair the ATPase cycle and knowing at least from the ΔN mutant that overall structure and localization to Morc3 binding sites are not impaired, we conclude that a functional ATPase cycle is necessary for Morc3 functionality. In addition, we have several experiments indicating SUMO-dependent interaction with Daxx.

6. The fact that 489 interacting proteins were found in the mass-spec analysis suggest that there might be a problem with false positives. Validation of some of these interactions (at least between Morc3 and Trim28, Setdb1, Daxx) with an alternative technique, such as a classic antibody-based IP, or with purified components, is essential.

To assess the protein context in which Morc3 operates, we used crosslinking ChIP-MS. This approach will capture stable and transient direct interactions with Morc3. However, ChIP-MS will also capture indirect interactions, dependent on the crosslinking/shearing conditions. Therefore, we would not argue that all 489 proteins are interaction partners of Morc3. We use the term "associate with

Morc3” in the manuscript to not generate the impression that these proteins are considered direct interaction partners.

To compare ChIP-MS with classical Immunoprecipitation we focused on validating our major finding, the interaction between Morc3 and Daxx (new Figure 6D,F). We could validate the interaction of Morc3 and Daxx by classical coIP. We detect loss of Daxx interaction with Morc3 mutant proteins and we could validate loss of Morc3 interaction with the Daxx dSIM mutant.

We see that some differences exist between ChIP-MS vs coIP. For example, ChIP-MS of Morc3 CW mutant only showed slightly impaired Daxx interaction, whereas this interaction is strongly affected in coIP. It is possible that the Morc3 CW mutant displays a very unstable interaction with Daxx that can be captured upon crosslinking, but which is not stable enough to be captured by coIP.

7. ChIP-seq data of H3.3 in Morc3-wt/KO cells would be very valuable in order to interpret the importance of this potential interaction between Morc3/Daxx/H3.3

We performed a comprehensive H3.3 ChIP-seq analysis for wild type, Morc3 ko, Morc3 wild type and mutant rescue cell lines (new Figure 7, Supplementary Figure S13,S14). We detect H3.3 association with Morc3 binding sites and Morc3 bound ERVs. Morc3 ko cells display strongly reduced H3.3 levels on Morc3 binding sites and ERVs. Morc3 rescue cells re-establish H3.3 patterns. Morc3 mutant rescue cell lines fail to re-establish H3.3, consistent with failed Daxx interaction of these mutant proteins.

8. Are the same ERV-integrants activated in H3.3-KO, Daxx-KO and Morc3-KO cells?

We performed a bioinformatic analysis of gene and ERV transcription in H3.3, Daxx and Morc3 ko cells using our dataset for Morc3 and published mutant vs rescue data for Daxx and H3.3 (Hoelper 2017). We found a subset of Morc3 regulated ERVs with transcriptional up-regulation in Daxx and H3.3 ko cells. Some Morc3 target genes are also de-repressed in Daxx ko and H3.3 ko. These data support the notion that Daxx and H3.3 are in the same silencing pathway as Morc3. However, there are clear differences between the three ko cell lines that can be explained by (1) different design of the study (wt vs. ko for Morc3 and ko vs rescue for Daxx and H3.3); (2) different RNA-seq library generation and sequencing mode (50PE vs 100SE); (3) Morc3-independent roles of Daxx and H3.3.

9. The discussion is highly speculative regarding the interaction between Morc3 and Daxx. This part of the manuscript should be modified to better represent the presented data.

We changed the discussion a lot to be less speculative and to better represent our data.

10. The ChIP-seq data are in conflict with published data (Li et al 2016) showing association with H3K4me3 over promoter regions. In that case, a Morc3 antibody was used, rather than a knock-in strategy. Could the authors try to resolve the apparent discrepancy? There is also convincing structural and biophysical data using purified components from multiple labs that Morc3 binds to unmodified H3 and H3K4me3 but NOT H3K9me3. Indeed, in Li et al, H3K9me3 was a negative control since ‘it was anti-correlated with morc3 in the genome’. A manual inspection of the chip-seq on morc3 looks ok on first glance. But inclusion of supp figures with controls (e.g. show the input track, show their other negative controls e.g. the anti-FLAG track in parent cells, to check for non-specific ChIP) would be valuable.

We have analyzed H3K4me3 together with Morc3 localization to promoters (new Supplementary Figure S9). We did not detect promoter association of the knock-in allele. However, we could detect Morc3 promoter binding in the wild type rescue and several mutant rescue cell lines. In this context it is interesting to note that the Morc3 CW mutant does not show promoter association, in agreement with H3K4me3-based recruitment. Why the rescue proteins display promoter association is a bit unclear, but it could be linked with somewhat higher expression level, which is not as controlled as for the knock-in allele. Nevertheless, we would agree with the view that Morc3 can associate with promoters. Since H3K4me3 is not changed in Morc3 ko and, expression changes of Morc3 target genes can be explained by proximal (ERV) enhancer activation, we think that Morc3 promoter binding has a minor impact on chromatin or transcriptional regulation. Therefore, we decided to keep the focus on the ERV binding sites, where we detect clear Morc3 dependent chromatin changes.

11. The authors should compare the results of their screen to a similar recent screen in mESCs (Bourc'his lab, <https://www.biorxiv.org/content/10.1101/2020.03.24.005488v1>)

We compared our screen with Chelmicki et al and found only very limited overlap (new Supplementary Figure S1). Importantly, major silencing factors were identified in both screens (Atrx, Daxx, Trim28, Dnmt1, Uhrf1) and also Morc3 was top scoring in both screens. We think the differences in the silencing inducing sequences and overall reporter design could explain the limited overlap.

12 Is it correct that nk the rescue rna-seq only has 1 replicate? If so, please perform additional

We have three replicates for the rescue RNA-seq analysis (see Supplementary Table 9).

13 The authors should state that the setdb1 chip is from a knock in

We added this information in the figure legend of Figure 2.

14. It is unclear where the trim28 chip-seq comes from

The data comes from Yang et al., 2015. We added this information in the figure legend of Figure 2.

Reviewer #2 (Remarks to the Author):

Groh et al.

This paper reports the finding, through a genome-wide CRISPR KO screen, that the Morc3 gene is important in silencing specific families of the ERVs in ES cells. Morc3 KO cells show reduced silencing, altered chromatin marks. Wild-type Morc3 restored silencing, but ATPase mutants did not. The paper shows that Morc3 interacts with the H3.3 chaparone Daxx, and that this required ATP binding, and Morc3 SUMOylation. Finally, Morc3 KO cells show reduced H3.3 at Morc3 binding sites.

The target locus used for the screen is a short element in the Gag region of the IAP family of ERVs, one previously identified by the lab (as a SHIN, short heterchromatin inducing sequence), which is used to regulate a dox-inducible promoter of reporter gene expression. Hits were validated for both maintenance of silencing and establishment of new silencing, leading to Morc3 (among other hits).

Chip-seq showed that Morc3 was localized to many ERVs, overlapping with sites of Setdb1 and Trim28 binding. Morc3 KO ES cells showed upregulation of many genes, three of which are noted to be near Morc3 binding sites. They also showed upregulation of some ERVs, though not as strongly as Setdb1 KO. There were complex changes in H3K9me3 patterns at Morc3 binding sites. ATAC seq revealed hundreds of sites with increased accessibility in the KO cells.

Proteomic analysis of ChIP samples revealed co-IP of Morc3 with hundreds of proteins, including Daxx, which bound to wt Morc3 but not to an ATP binding site mutant. Daxx is a known heterochromatin silencer, working through a SUMO interaction motif, to mediate H3.3 incorporation. H3.3 ChIP revealed overlap with Morc3 sites, and reduced signal at many sites in Morc3 KO cells.

The model is presented that normally there is ongoing, rapid turnover of nucleosomes in silent heterochromatin, and that Daxx is needed to re-establish H3.3 and maintain silencing, and that this requires Daxx interaction with SUMOylated Morc3. An active ATPase cycle is also proposed to be needed for Morc3 function.

This is a very nice paper with an impressive amount of new information about ERV silencing. Morc1 and Morc2 have been shown to be important for silencing of ERVs in other settings, and here this extended to Morc3; and we also learn of its function through Daxx and H3.3 and H3K9me3. This is close to a perfect paper: the study is well-designed, the experiments are strong, the data are all solid, and the paper is clear and well-written. There is a huge amount of work presented, and there is much new information that will be of interest to the community.

I can find nothing significant to suggest. Perhaps a few comments on the history of the SHIN site would be helpful.

We are very thankful to the reviewer for this highly positive evaluation of our work. We have added information regarding the SHIN sequence in the introduction.

Reviewer #3 (Remarks to the Author):

In this manuscript, Groh et al identify the ATPase Morc3 as a novel protein involved in ERV silencing in mouse embryonic stem cells. Loss of Morc3 leads to loss of H3K9me3 and increased chromatin accessibility at a subset of ERVs. The authors then study the function of three established Morc3 mutants. They find that (1) an ATP hydrolysis mutant cannot bind to chromatin, and (2) an ATP binding mutant cannot bind to DAXX, a protein previously identified by several groups to be involved in ERV silencing. Interaction between Morc3 and DAXX requires the SIM domain of the latter. The model put forth is that Morc3-DAXX binding is required for DAXX recruitment to ERVs, H3.3 deposition, and subsequent chromatinization and heterochromatinization.

The validation of Morc3 as an ERV silencing protein using the reporter plasmid and stable reporter ESCs are clean and convincing. The ChIP-seq data localizing Morc3 to ERVs is also straightforward and in agreement with the screen validation. I have some minor concerns about the integration of various omics data sets in Figures 3 and 4. I have more serious concerns about how well claims from Figures 5 and 6 have been substantiated to support the model put forth in Figure 7. Should the

authors provide more evidence in support of this model, I think this work will make an important contribution towards our understanding of heterochromatin formation at ERVs.

Suggested improvements in computational analysis:

1. The authors suggest that Morc3 is suppressing LTRs which may otherwise act as enhancers for nearby genes. Do these regions gain marks of active enhancers such as p300 binding and H3K27ac?

We thank the reviewer for this suggestion. We performed H3K27ac ChIP-seq experiments in wild type, Morc3 ko and Morc3 rescue cells (new Figure 4D,E, Supplementary Figure S5). Interestingly, we detect a subset of Morc3 peaks which clearly gain H3K27ac. A small subset of these peaks also displays increased chromatin accessibility. Genomic feature analysis revealed that the Morc3 binding sites with H3K27ac gain relate to (1) ERVs (in particular LTRIS2) and (2) ENCODE CREs (developmental enhancers). Based on our analysis we now think that Morc3 helps to suppress distinct developmental enhancers, in addition to its role in ERV regulation.

2. The authors identify 64 upregulated genes with Morc3 peaks <100kb. From Fig. 3B, it looks as though the authors have RNA-seq from rescue ESCs. This would allow the authors to provide a more comprehensive assessment of whether Morc3 OE can rescue (1) all global transcription defects observed in the KO, or perhaps (2) a subset of DE based on proximity of a Morc3 binding site to the promoter as delineated in Supplemental Table 5. This would be preferable over the select RT-PCR shown in Fig. 3C.

We now provide a deeper analysis of transcriptional changes in Morc3 rescue cells (new Supplementary Figure S4). We detect that both gene and ERV expression are largely restored in Morc3 rescue cell lines.

3. I'd like to see some statistics associated with the H3K9me3 ChIP-seq data in Figure 4A. The hierarchical clustering and heat map that has been performed does not allow this.

We addressed this issue by quantifying H3K9me3 coverage on the center of all Morc3 binding sites (new Figure 4B). These data were subjected to ANOVA analysis to assess differences between wt, ko and rescue. We detect significant reduction in Morc3 ko. There is no significant difference between wt and rescue, in agreement with a good rescue effect.

4. It's interesting that Morc3-bound sites show differential effects on H3K9me3 when Morc3 is lost. This does not appear related to the level of Morc3 enrichment within a particular group. Is there any other genomic feature that influences how sensitive a particular region is to Morc3 loss?

We assessed genomic features of Morc3 binding sites in the different clusters (new Supplementary Figure S5A). We could not detect anything specific that could explain these differences.

5. The directionality of the H3K9me3 loss in Morc3 KO cells is quite interesting and I'd be curious to know more about the mechanism behind this.

We are also very curious about the mechanisms behind. Somehow boundaries close to some Morc3 binding sites must exist to maintain the separation between active and inactive chromatin in close vicinity. In follow-up studies, we may dig deeper into this effect, but at this point we cannot explain it.

6. The ATAC-seq data are poorly linked back to Morc3 binding. The authors anecdotally claim that differentially regulated ATAC peaks are generally within proximity of a Morc3 peak, but a few browser tracks are not enough to make this claim and this point must be fully addressed computationally.

a. What is the distribution of distance between Morc3 peaks and ATAC peaks?

b. Do all Morc3-bound peaks become more open in the absence of Morc3 and is the difference statistically relevant?

c. Is differential accessibility correlated with decreased H3K9me3?

Overall, while Figure 4D is quite striking, the earlier data in this figure lack appropriate depth of analysis and comparative analysis.

We performed a new data integration analysis, also considering the H3K27ac results (new Figure 4E). Our interpretation is as follows: Morc3 knock-out results in increased accessibility on a subset of bindings sites, largely overlapping with H3K9me3 loss and H3K27ac gain. Most Morc3 peaks do not show higher chromatin accessibility, even though they may lose H3K9me3. The reason is most likely that regions, where Morc3 acts in competition with transcriptional activators, loss of Morc3 more easily allows establishment of more accessible, active chromatin. In most regions, active chromatin is not established upon loss of Morc3, maybe because transcriptional activators for these regions are not present, or redundancy with other pathways keeps these regions inaccessible. Differential ATAC peaks in the vicinity of Morc3 reflect enhancer activation (see example screenshots for Cd200, Ube2l6, Irak3). In addition, we now specifically mark Morc3 binding sites in the genome-wide ATAC-seq analysis (all ATAC-seq dot plots).

7. The authors have performed ATAC-seq using their Morc3 mutants (Fig. 5E and Supplemental Fig. 6), but they do not present this genomic analysis. Again, browser track snapshots of a few regions are not sufficient when genome-wide analyses can be performed.

We have included a bioinformatic analysis of genome-wide ATAC-seq changes in these mutants (new Supplementary Figure S10D). In all mutants and in agreement with their loss-of-function phenotypes, we detect very similar ATAC changes as compared to Morc3 knock-out cells.

Other major points:

8. It's really critical that the authors address head on that the reported ligand for the CW domain is H3K4me3. The intro to this section is difficult to understand because the authors seem not to want to state this. While they do bring up this point in the discussion, I don't feel that it has been sufficiently addressed. Based on Fig. 2A there is very little Morc3 binding at promoters (sites of H3K4me3 enrichment) and the authors state that they do not observe H3K4me3 enrichment in ERV heterochromatin. It's difficult to understand why the CW domain should be required for their SHIN silencing assay or for transcription regulation as shown in Figure 5C and 5D. Morc3 localization seems least affected by the CW mutation, but the authors do not provide a satisfying explanation as to why this region is important for ERV silencing.

We have re-examined Morc3 promoter localization (new Supplementary Figure S9). In Morc3 rescue cell lines we do detect promoter association of Morc3. The Morc3 CW mutant fails to localize to promoters. These data support CW-mediated recruitment of Morc3 to H3K4me3 promoters.

However, we do not detect specific chromatin changes on promoter binding sites and transcriptional changes in Morc3 ko cells can rather be attributed to enhancer activation. Therefore, we think that promoter binding of Morc3 does not affect transcriptional regulation a lot.

What is the role of the CW domain in ERV regulation? It is quite clear that the CW domain is a negative regulator of Morc3 ATPase activity. In absence of ligand (H3K4me3), ATPase activity is reduced. Our analysis also of the other mutants demonstrated that the ATPase cycle is needed for Daxx interaction and H3.3 deposition. We would explain the effect of the CW mutant by reduced ATPase activity (resulting in reduced ATPase cycle) and impaired Daxx interaction. The ligand for the CW domain in ERV heterochromatin is not known. We do not at all exclude H3K4me3, it could be transient or at low levels. However, it could also be that the CW domain interacts with other (methylated) proteins.

9. In general, the proteomics analyses should be validated using other biochemical methods. The model is that Morc3 is a direct binding partner of DAXX mediated by Morc3 SUMOylation and the DAXX SIM domain, but the data at this stage do not fully support this model. If the site of SUMOylation on Morc3 is known, this would be a valuable mutant in this study.

We have performed validation experiments using coIP and included a new Morc3 mutant protein which cannot be SUMOylated (new Figure 6C,D,F). These data support the SUMO-dependent interaction between Morc3 and Daxx.

10. The model is that the entire ATP hydrolysis cycle of Morc3 is required for ERV silencing by regulating DAXX-mediated H3.3 deposition. This point needs additional experimental support. Given that its expression levels are low and the mutant does not bind to Morc3 target regions, I assume the ATP hydrolysis mutant acts as a Morc3 KO. However, the authors should more fully characterize the CW and ATP-binding mutants. It's quite curious that the CW mutant maintains all SETDB1-complex associations and is retained on chromatin yet results in SHIN reporter activation. What, then, is the function of the CW domain? Does CW mutant expression affect H3.3 deposition or H3K9me3 levels at ERVs? Further, the ATP-binding mutant binding to chromatin seems attenuated (see Supplemental Fig. 5B, 5C), yet this mutant maintains all SETDB1-complex associations (including ATRX) with the striking exception of DAXX. This result needs orthogonal validation. In addition, the same H3K9me3 and H3.3

genomic profiling should be completed from cells expressing the ATP.b mutant. The authors should also test whether each mutant affects nuclear DAXX levels. Ideally, DAXX ChIP should be provided as well to determine whether DAXX is lost from chromatin when Morc3 cannot bind ATP.

We have now done a comprehensive analysis of the Morc3 mutant rescue cell lines (Figures 5,7, Supplementary Figures S8, S10, S13) and included two additional mutants (ΔN – dimerization mutant, low ATPase activity; SUMOylation mutant). We profiled localization, ATAC-seq, H3K9me3 and H3.3 for all mutant cell lines. Although the ATP hydrolysis mutant does not localize to Morc3 binding sites, ATP binding mutant and ΔN mutant do show this localization. H3K9me3 and H3.3 deposition are impaired in these mutants. The SUMOylation mutant also localized to Morc3 peaks but fails to restore H3K9me3 or H3.3. In addition, our new coIP data (Figure 6D) show that all mutants display impaired Daxx interaction. Nuclear Daxx levels are unchanged (Supplementary Figure S12). Unfortunately, we could not establish a working ChIP-seq approach for Daxx.

However, based on our new data we think that our line of argumentation is strengthened. The Morc3 ATPase cycle and Morc3 SUMOylation are important for Daxx-mediated H3.3 deposition.

Minor Points:

1. I'm not sure why the authors state in Line 61 that no screens have been performed using an endogenous ERV sequence as they themselves published such an shRNA screen in their 2015 EMBO Rep paper.

We removed this statement also considering the recent screen by the Borc'his lab.

2. Validation of Morc3-3xFlag is lacking (i.e., genotyping, Flag western blot).

We added the validation western in Supplementary Figure S2.

3. Authors should provide a table to better document all sequencing data used in this study.

We have this table in the supplement (Supplementary Table 9).

4. There are some issues with the callouts for Figure 4.

We have corrected this.

5. Lines 207-210 are speculative. ATAC-seq does not measure whether TFs are bound, it measures whether chromatin is accessible. These lines in particular, but there are a number of concluding statements interspersed throughout the results that do not describe the results as presented and are speculative in nature. These would be better suited for the discussion.

We extensively changed results and discussion section and took this into consideration.

6. Supplemental Fig. 5C should be on the same scale. Both ATP binding mutant and ATP hydrolysis mutant look reduced at target sites. They also both look more lowly expressed compared to WT rescue.

We adjusted the scale, except for the Morc3 SUMOylation mutant which displayed very strong IAPEz association (now Supplementary Figure S8E).

REVIEWERS' COMMENTS

Reviewer #1 (Remarks to the Author):

The authors have done an impressive amount of work for the revision and responded to all comments in a satisfactory way. I have no further comments at this stage. The manuscript is ready for publication.

Reviewer #3 (Remarks to the Author):

I'd like to congratulate the authors on a very interesting study. I have no further comment before publication. Nice work!